# MicroRNA cluster miR-17-92 regulates multiple functionally related voltage-gated potassium channels in chronic neuropathic pain

Atsushi Sakai[1], Fumihito Saitow[1], Motoyo Maruyama[1,2], Noriko Miyake[3], Koichi Miyake[3], Takashi Shimada[3], Takashi Okada[3] & Hidenori Suzuki[1]

miR-17-92 is a microRNA cluster with six distinct members. Here, we show that the miR-17-92 cluster and its individual members modulate chronic neuropathic pain. All cluster members are persistently upregulated in primary sensory neurons after nerve injury. Overexpression of miR-18a, miR-19a, miR-19b and miR-92a cluster members elicits mechanical allodynia in rats, while their blockade alleviates mechanical allodynia in a rat model of neuropathic pain. Plausible targets for the miR-17-92 cluster include genes encoding numerous voltage-gated potassium channels and their modulatory subunits. Single-cell analysis reveals extensive co-expression of miR-17-92 cluster and its predicted targets in primary sensory neurons. miR-17-92 downregulates the expression of potassium channels, and reduced outward potassium currents, in particular A-type currents. Combined application of potassium channel modulators synergistically alleviates mechanical allodynia induced by nerve injury or miR-17-92 overexpression. miR-17-92 cluster appears to cooperatively regulate the function of multiple voltage-gated potassium channel subunits, perpetuating mechanical allodynia.

[1] Department of Pharmacology, Nippon Medical School, 1-1-5 Sendagi, Bunkyo-ku, Tokyo 113-8602, Japan. [2] Division of Laboratory Animal Science, Nippon Medical School, 1-1-5 Sendagi, Bunkyo-ku, Tokyo 113-8602, Japan. [3] Department of Biochemistry and Molecular Biology, Nippon Medical School, 1-1-5 Sendagi, Bunkyo-ku, Tokyo 113-8602, Japan. Correspondence and requests for materials should be addressed to A.S. (email: sa19@nms.ac.jp).

A microRNA (miRNA) cluster is a polycistronic gene in which several miRNAs are encoded in a single primary, or nascent, transcript. Small non-coding mature miRNAs are produced by stepwise cleavage of primary miRNA transcribed from the genome, and inhibit expression of diverse genes through a sequence-dependent binding to a specific 3′-untranslated region (UTR) sequence in mRNAs. About 40% of microRNAs are estimated to form clusters whose physiological importance is largely unknown[1], although the roles of individual miRNAs in a variety of physiological and pathological states are increasingly being recognized.

The miR-17-92 cluster encodes six distinct miRNAs in a single primary transcript (miR-17, miR-18a, miR-19a, miR-19b, miR-20a and miR-92a) compared with common miRNA clusters that comprise only two or three miRNAs[1]. The miR-17-92 null mutation is lethal because of lung hypoplasia and ventricular septal defects[2], and miR-17-92 dysregulation has been observed in diverse neurological diseases such as autism spectrum disorder and Alzheimer's disease[3–5]. Our previous microarray analysis[6] showed that miR-17-92 cluster members were upregulated in the dorsal root ganglion (DRG) after nerve injury. The pathophysiological significance of miRNA cluster in neurological disorders is not yet understood.

Voltage-gated potassium channels are critical regulators of neuronal excitability, acting by modulating action potential generation, firing rate or neurotransmitter release[7]. Voltage-gated potassium channels are encoded by ∼40 genes: six encode the $K_V1.4$, $K_V3.3$, $K_V3.4$ and $K_V4.1$–4.3 channel subtypes that mediate fast-inactivating A-type currents, while the others are delayed rectifiers[8]. Expression of a variety of voltage-gated potassium channels—and thus both currents—are consistently decreased in DRG neurons after nerve injury, making potassium channels attractive targets for the treatment of neuropathic pain[7]. Neuropathic pain is frequently caused by lesions or disease of primary sensory neurons; their inputs to the spinal cord are critical for the development and maintenance of chronic pain[9]. miRNAs reportedly regulate gene translation in neuropathic pain states[10,11]. Although the molecular mechanisms regulating the expression of several potassium channels have recently been illuminated[12–15], the pathophysiological mechanism by which potassium channel expression is coordinately downregulated in DRG neurons is incompletely understood.

Here, we show the combinatorial impact of miR-17-92 cluster members on chronic neuropathic pain through regulation of functionally related multiple voltage-gated potassium channels and their modulatory subunits, especially those responsible for the A-type potassium current.

## Results

**miR-17-92 cluster is upregulated in DRG neurons after injury.** Using quantitative PCR (qPCR), we first confirmed that expression of each miR-17-92 cluster member (Supplementary Fig. 1a) was significantly upregulated in the DRG 14 days after ligation of the fifth lumbar (L5) nerve, as observed in our previous microarray analysis[6] (Fig. 1a). Upregulation was sustained from day 1 to day 28 after L5 spinal nerve ligation (SNL; Fig. 1a); neuropathic pain was also evident at the same time (Fig. 1b). Next, we examined the expression of pri-miR-17-92 (the primary transcript of miR-17-92 cluster), finding that it was also upregulated from days 1 to 28 after SNL (Fig. 1c), suggesting that the upregulation of mature miR-17-92 cluster miRNA expression reflected transcriptional upregulation of the miR-17-92 cluster. In contrast, expression of miR-17-363 (miR-106a-363 homologue in humans), a miR-17-92 paralog encoding similar mature miRNAs (Supplementary Fig. 1a), was not consistently

elevated (Supplementary Fig. 1b). miR-17-92 was also upregulated in the DRG neurons in another neuropathic pain model, spared nerve injury (Supplementary Fig. 2a). In contrast, miR-17-92 expression was unchanged in the injury-spared L4 DRG and L5 dorsal spinal cord ipsilateral to the L5 SNL (Supplementary Fig. 2b,c). Expression of miR-17-92 cluster members was not elevated in the L5 DRG in rats with inflammatory pain of the hind paw provoked by complete Freund's adjuvant (CFA) (Supplementary Fig. 2d), despite the partial overlap of the molecular mechanisms of inflammatory and neuropathic pain and despite CFA inducing comparable mechanical allodynia (Supplementary Fig. 2e). These results suggest nerve injury-specific involvement of miR-17-92 in pain behaviours.

**miR-17-92 overexpression causes mechanical allodynia.** To examine the role of miR-17-92 in pain behaviours, we induced expression of the whole miR-17-92 cluster specifically in the L5 DRG neurons of rats by local injection of an adeno-associated virus (AAV) vector. An AAV vector encoding the whole miR-17-92 cluster was successfully transduced into DRG neurons of all cell sizes, as evidenced by enhanced green fluorescent protein (EGFP) immunofluorescence (Fig. 2a and Supplementary Fig. 3a,b), as previously described[16]. Expression of miR-17-92 primary transcript was increased 7 days after AAV vector injection (Fig. 2b). Significant mechanical allodynia, but not thermal hyperalgesia, was observed in rats with miR-17-92 overexpression 7 days after AAV vector injection (Fig. 2c,d).

As the miR-17-92 cluster encodes six distinct miRNAs, we attempted to identify the cluster members responsible for mechanical allodynia. An AAV vector expressing each individual cluster member was injected into the L5 DRG; 7 days later increased expression of each corresponding miRNA was confirmed (Supplementary Fig. 3c). Significant mechanical allodynia was observed in rats overexpressing miR-18a, miR-19a, miR-19b or miR-92a, but not in those overexpressing miR-17 or miR-20a (Fig. 2e). Thermal hyperalgesia was not induced by overexpression of any of the individual cluster members (Supplementary Fig. 4).

**Blockade of miR-17-92 members alleviates neuropathic pain.** We further examined the therapeutic potential of miRNA blockade for neuropathic pain using tough decoy (TuD) anti-sense RNA, an efficient and specific inhibitor of miRNA that prevents it from binding to its target mRNA[17]. Only the cluster members whose overexpression reduced the mechanical paw withdrawal threshold (miR-18a, miR-19a, miR-19b and miR-92a) were examined. The efficacy of each of the antisense RNAs was confirmed using a luciferase assay (Supplementary Fig. 5). An AAV vector encoding each antisense RNA was injected into the L5 DRG 7 days before SNL. None of the antisense RNAs affected the mechanical threshold (Fig. 3a) or thermal latency (Supplementary Fig. 6a) in intact rats at day 0 (before SNL). Injection of AAV vector encoding miR-18a, miR-19a, miR-19b or miR-92a antisense RNA significantly prevented mechanical allodynia compared with control antisense RNA (Fig. 3a). Furthermore, the established mechanical allodynia was reversed by injection of a mixture of AAV vectors encoding antisense RNAs against miR-18a, miR-19a, miR-19b and miR-92a 7 days after SNL (Fig. 3b). Thermal hyperalgesia was not affected by the antisense RNAs (Supplementary Fig. 6a). Interestingly, miR-92a antisense RNA also significantly suppressed thermal hyperalgesia, to a lesser extent (Supplementary Fig. 6a). miR-17-92 inhibition did not affect motor function as assessed by open field and rotarod tests (Supplementary Fig. 6b,c). We further analysed the effect of miR-17-92 on spontaneous pain in

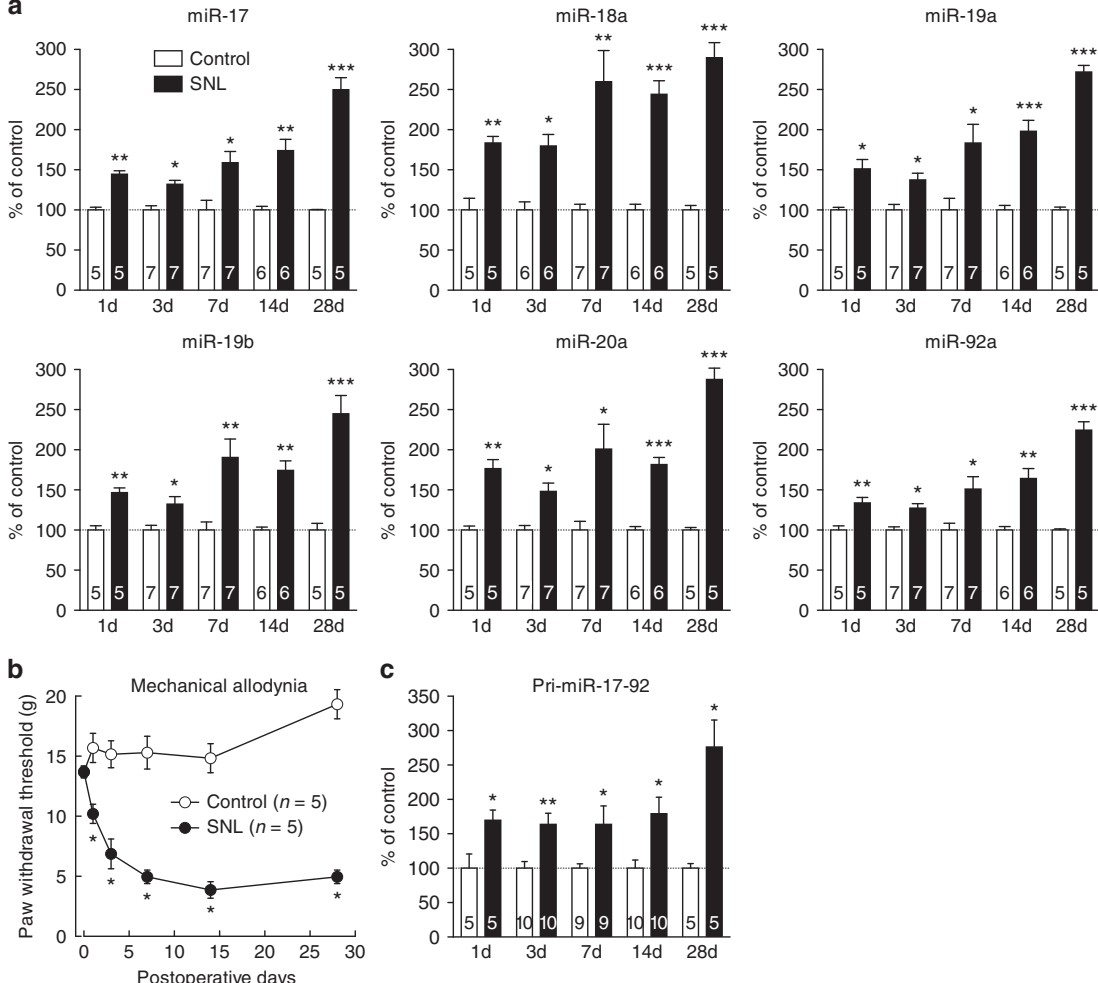

**Figure 1 | miR-17-92 cluster upregulation in L5 DRG neurons after nerve injury. (a,c)** Change in expression of mature miRNAs of the miR-17-92 cluster (**a**) and primary transcripts of miR-17-92 (**c**) in the L5 DRG over time after SNL. Numbers of samples are indicated at the base of each bar. *$P < 0.05$, **$P < 0.01$ and ***$P < 0.001$ (miR-17, $P = 0.002$ for 1d, $P = 0.010$ for 3d, $P = 0.017$ for 7d, $P = 0.004$ for 14d and $P < 0.001$ for 28d; miR-18a, $P = 0.006$ for 1d, $P = 0.011$ for 3d, $P = 0.005$ for 7d, $P < 0.001$ for 14d and $P < 0.001$ for 28d; miR-19a, $P = 0.017$ for 1d, $P = 0.033$ for 3d, $P = 0.022$ for 7d, $P < 0.001$ for 14d and $P < 0.001$ for 28d; miR-19b, $P = 0.002$ for 1d, $P = 0.049$ for 3d, $P = 0.005$ for 7d, $P = 0.002$ for 14d and $P < 0.001$ for 28d; miR-20a, $P = 0.008$ for 1d, $P = 0.015$ for 3d, $P = 0.020$ for 7d, $P < 0.001$ for 14d and $P < 0.001$ for 28d; miR-92a, $P = 0.006$ for 1d, $P = 0.016$ for 3d, $P = 0.011$ for 7d, $P = 0.002$ for 14d and $P < 0.001$ for 28d; pri-miR-17-92, $P = 0.043$ for 1d, $P = 0.005$ for 3d, $P = 0.048$ for 7d, $P = 0.022$ for 14d and $P = 0.011$ for 28d) compared with the contralateral intact side, paired $t$-test. (**b**) Paw withdrawal responses to mechanical stimuli evaluated on the SNL and contralateral sides. Error bars are s.e.m. *$P < 0.05$ ($P = 0.043$ for 3d, $P = 0.042$ for 7d, $P = 0.041$ for 14d and $P = 0.039$ for 28d) compared with the contralateral side, Mann–Whitney $U$ test.

a combined neuropathic and inflammatory pain model, in which obvious spontaneous paw liftings represent possible spontaneous pain[18]. miR-17-92 inhibition did not significantly decrease the spontaneous paw liftings, although it had a tendency to suppress spontaneous pain behaviour (Supplementary Fig. 6d).

**Bioinformatic analysis of miR-17-92 cluster targets.** To illuminate the mechanisms underlying miR-17-92-mediated mechanical allodynia, we searched target genes for miR-17-92 cluster members that could be responsible for pain behaviour using Ingenuity Pathway Analysis (IPA; Qiagen K.K., Tokyo, Japan). The number of putative target genes for miR-18a, miR-19a/b (miR-19a and miR-19b have the same seed sequence) and miR-92a were 695, 1,448 and 1,138, respectively (a total of 2,834 genes). We further restricted the potential target genes based on the mRNA expression profile 28 days after L5 SNL, as persistent miRNA-target mRNA complexing can lead to mRNA decay[19]. Comprehensive microarray analysis of mRNA expression

changes[20] available in Gene Expression Omnibus (www.ncbi.nlm.nih.gov/geo/; GEO accession GSE24982) identified 1,470 significantly downregulated mRNAs in the DRG 28 days after L5 SNL. Intriguingly, 20.3% of the downregulated genes (299 genes) were predicted as targets of pain-relevant miR-17-92 cluster members (Supplementary Fig. 7a and Supplementary Data 1), suggesting a broad modulatory role for miR-17-92 cluster following nerve injury. These plausible target genes were then analysed using IPA to explore their associated functions. Downstream effects analysis, which predicts the likelihood of involvement in several downstream biological processes, identified 'neurological disease' as the most significant (Supplementary Fig. 7b), supporting a functional role for miR-17-92 cluster in the modulation of gene expression after SNL. Network analysis also showed that plausible target genes comprised a relatively high proportion of genes associated with the network function of cell-to-cell signalling and interaction, nervous system development and function, and neurological disease (Supplementary Fig. 7c,d).

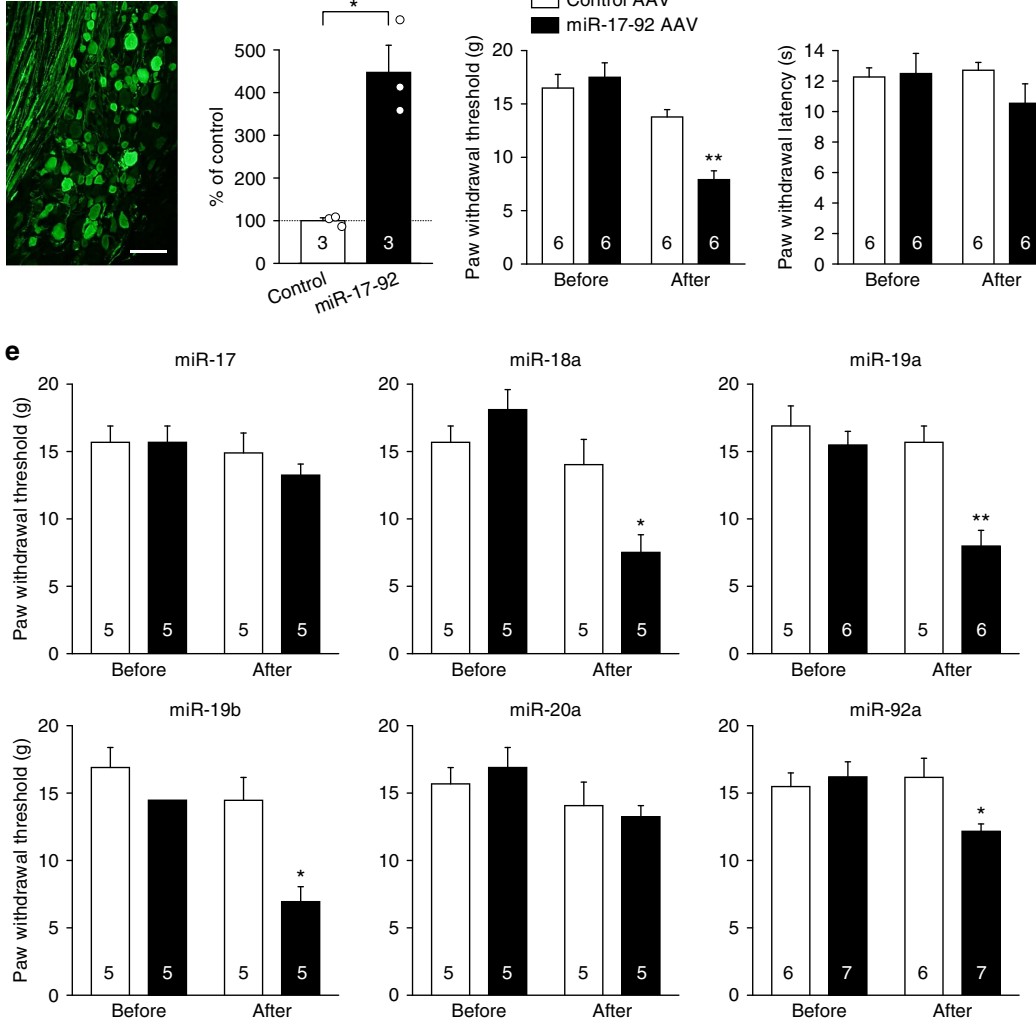

**Figure 2 | miR-17-92 cluster overexpression in DRG neurons causes mechanical allodynia.** (**a**) Representative image of EGFP immunofluorescence in the L5 DRG 7 days after injection of the AAV vector encoding miR-17-92 and EGFP. Scale bar, 100 µm. (**b**) Expression levels of miR-17-92 primary transcript in L5 DRGs 7 days after control or miR-17-92 AAV vector injection. Numbers of samples are indicated at the base of each bar. *$P = 0.031$ by Welch's test. (**c–e**) Paw withdrawal threshold (**c,e**) and latencies (**d**) to mechanical and thermal stimuli, respectively, were evaluated on the injected sides before and 7 days after the injection of AAV vector encoding the whole miR-17-92 cluster (**c,d**) or individual miR-17-92 cluster members (**e**). Error bars are s.e.m. *$P < 0.05$ and **$P < 0.01$ ($P = 0.004$ for miR-17-92; $P = 0.023$ for miR-18a; $P = 0.004$ for miR-19a; $P = 0.008$ for miR-19b; $P = 0.014$ for miR-92) compared with the control injection, Mann–Whitney $U$ test.

**miR-17-92 downregulates voltage-gated potassium channels.** Many voltage-gated potassium channel α subunits were among the plausible target genes downregulated following nerve injury and predicted as targets of pain-relevant miR-17-92 cluster members (Fig. 4a), including $K_V1.1$, $K_V1.4$ and $K_V4.3$ (refs 21–25). In addition, positive modulators of the voltage-gated potassium channels DPP10 and $Na_V\beta1$ (refs 26–28) were also predicted miRNA targets. Therefore, we examined whether voltage-gated potassium channel subunits are targeted by miR-17-92 cluster miRNAs using a luciferase assay. As an miRNA generally recognizes the 3′-UTR of mRNAs[19], each 3′-UTR sequence of a candidate target gene was inserted downstream of the firefly luciferase gene in a plasmid vector. Activities of luciferases with putative target 3′-UTR sequences were generally decreased by their corresponding miRNAs (Fig. 4b), although *Kcna1* and *Dpp10* 3′-UTRs were not targeted by miR-92a or miR-18a, respectively. Prior transfection of TuD miRNA antisense RNAs inhibited the decrease in luciferase activity by corresponding miRNAs (Supplementary Fig. 5), validating the

effectiveness of the antisense RNAs. To confirm that the predicted seed sequences were responsible for miRNA-mediated luciferase suppression, these seed sequences were mutated to mismatch the miRNA sequences (Supplementary Fig. 8). As miR-19a/b and miR-92a have two predicted binding sites for *Kcna4* 3′-UTR and *Kcnc4* 3′-UTRs, respectively, both sites were mutated. Activities of luciferases with mutated 3′-UTRs were no longer suppressed by corresponding miRNAs (Fig. 4b), indicating that the predicted seed sequences were directly targeted in a sequence-specific manner by corresponding miRNAs. Importantly, all seed sequences are conserved among mammals (Supplementary Fig. 8), highlighting the potential importance of potassium channel modulation by miR-17-92.

Next, we investigated the role of miR-17-92 in the modulation of potassium channel subunit expression *in vivo*. miR-17-92-expressing cell types and potassium channel expression patterns were examined using laser microdissection followed by reverse transcription PCR (RT-PCR). Pri-miR-17-92 expression in laser-captured single DRG neurons (Fig. 5a,b) was positively detected

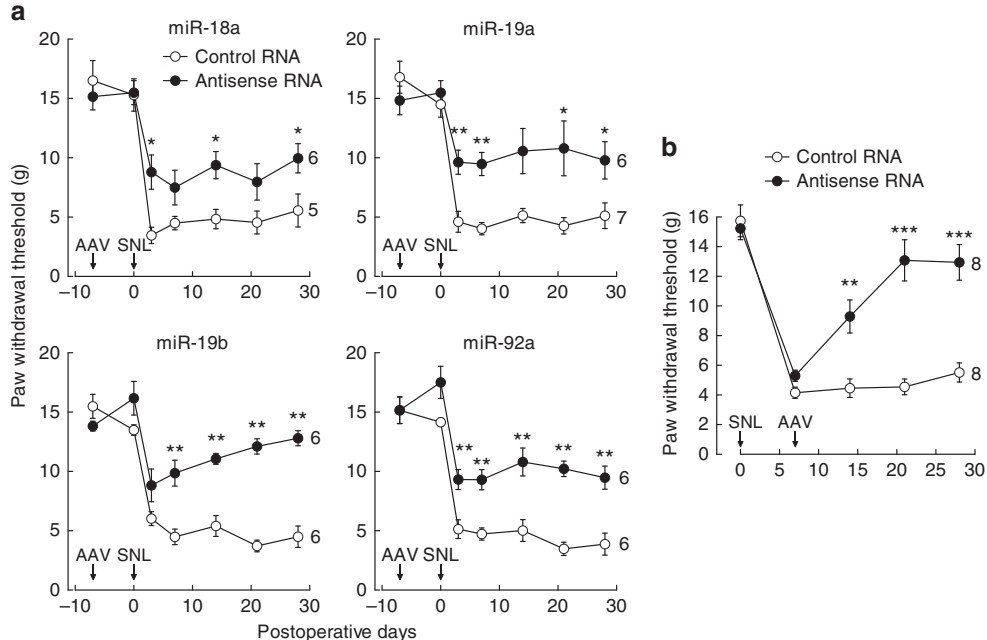

**Figure 3 | Blockade of miR-17-92 cluster members suppresses mechanical allodynia in SNL.** Paw withdrawal responses to mechanical stimuli were evaluated on the AAV-injected/SNL side. Numbers of animals are indicated to the right of the plots. (**a**) AAV vector encoding control TuD antisense RNA or TuD antisense RNA against each miR-17-92 cluster member was administered to the L5 DRG 7 days before SNL (indicated by arrows). (**b**) Mixture of AAV vectors encoding TuD antisense RNAs against miR-18a, miR-19a, miR-19b and miR-92a was administered to the L5 DRG 7 days after SNL (indicated by arrows). Error bars are s.e.m. *$P<0.05$, **$P<0.01$ and ***$P<0.001$ (miR-18a, $P=0.011$ for 3d, $P=0.030$ for 14d and $P=0.030$ for 28d; miR-19a, $P=0.008$ for 3d, $P=0.002$ for 7d; $P=0.014$ for 21d and $P=0.035$ for 28d; miR-19b, $P=0.004$ for 7d, $P=0.002$ for 14d, $P=0.002$ for 21d and $P=0.002$ for 28d; miR-92a, $P=0.002$ for 3d, $P=0.002$ for 7d, $P=0.009$ for 14d, $P=0.002$ for 21d and $P=0.004$ for 28d; antisense RNA mixture, $P=0.003$ for 14d, $P<0.001$ for 21d and $P<0.001$ for 28d) compared with the control injection, Mann–Whitney $U$ test.

in small DRG neurons of cell size $<600\,\mu m^2$ and medium/large neurons of cell size $>600\,\mu m^2$ (Fig. 5c). Small and large DRG neurons represent putative C-fibre and A-fibre neurons, respectively. RT-PCR also detected a massive overlap of potassium channel subunit expression in DRG neurons (Fig. 5c). Importantly, voltage-gated potassium channel subunits were extensively detected in DRG neurons positive for the expression of miR-17-92. To further address the cell types expressing miR-17-92, TAC1 and P2X3 expressions were examined as peptidergic and non-peptidergic DRG neuronal markers, respectively. miR-17-92 was primarily localized to P2X3-expressing non-peptidergic DRG neurons, although it was also expressed in the TAC1-positive peptidergic DRG neurons (Fig. 5c). Then, we analysed the expression of potassium channels in the L5 DRG of rats 7 days after AAV vector injection. Microarray analysis revealed that injection of an AAV vector encoding the whole miR-17-92 cluster decreased the expression of all voltage-gated potassium channel subunits targeted by miR-17-92 ($n=4$; Supplementary Table 1). miR-17-92 overexpression also modulated the expression of other potassium channels (Supplementary Table 1) and non-potassium channel genes (Supplementary Data 2). Many of these genes that were not predicted as direct targets of miR-17-92 may also be indirectly modulated through the downregulation of miR-17-92 targets. qPCR confirmed the decreased expression of all voltage-gated potassium channel subunits targeted by miR-17-92 (Fig. 5d). Similarly, overexpression of each miR-17-92 cluster member decreased expression of corresponding voltage-gated potassium channel subunits (Fig. 5d), although $Na_V\beta1$ expression was mostly unaffected by miR-19a. Notably, the whole miR-17-92 cluster more robustly downregulated expression of potassium channel subunits targeted by several miRNAs than each cluster member alone. In contrast, other pain-related potassium channels ($K_V1.2$, $K_{ir}6.1$ and $BA_{CA}$)[7] that were not predicted as miR-17-92

targets did not show significant expression changes (Supplementary Fig. 9). Because $K_V1.1$, $K_V3.4$ and $K_V4.3$ are reportedly involved in mechanical allodynia, but not thermal hyperalgesia[22,29], these potassium channels appear to be particularly important for miR-17-92-mediated mechanical allodynia.

**miR-17-92 decreases potassium currents in DRG neurons.** Potassium channel α subunits targeted by miR-17-92 include three of six known potassium channels constituting rapidly inactivating potassium currents, or A-type currents (Fig. 4a). In addition, DPP10 and $Na_V\beta1$ subunits are shown to positively modulate $K_V4$ channels, which are major contributors to A-type currents[26,28], further suggesting the importance of miR-17-92 in the modulation of A-type potassium currents. Therefore, we performed whole-cell patch clamping of primary sensory neurons to measure A-type and non-A-type potassium currents. The non-A-type potassium current was recorded in the presence of the A-type potassium channel blocker, 3,4-diaminopyridine (DAP)[12]. The DAP-sensitive A-type potassium current was obtained by subtracting the non-A-type current from the total potassium currents. In L5 DRG neurons prepared from rats injected with AAV vector encoding the whole miR-17-92 cluster 7 days beforehand, total potassium currents were diminished in small DRG neurons of cell size $<600\,\mu m^2$ (Fig. 6a,c), consistent with the reduction observed in L5 DRG neurons following L5 SNL (Supplementary Fig. 10). Specifically, miR-17-92 (Fig. 6a,c) and SNL (Supplementary Fig. 10) markedly reduced A-type potassium currents, consistent with the preferential expression of target potassium channels mediating A-type currents in small DRG neurons (Fig. 4a and Fig. 5c). In contrast, miR-17-92 cluster only slightly reduced the non-A-type current (Fig. 6a,c), while

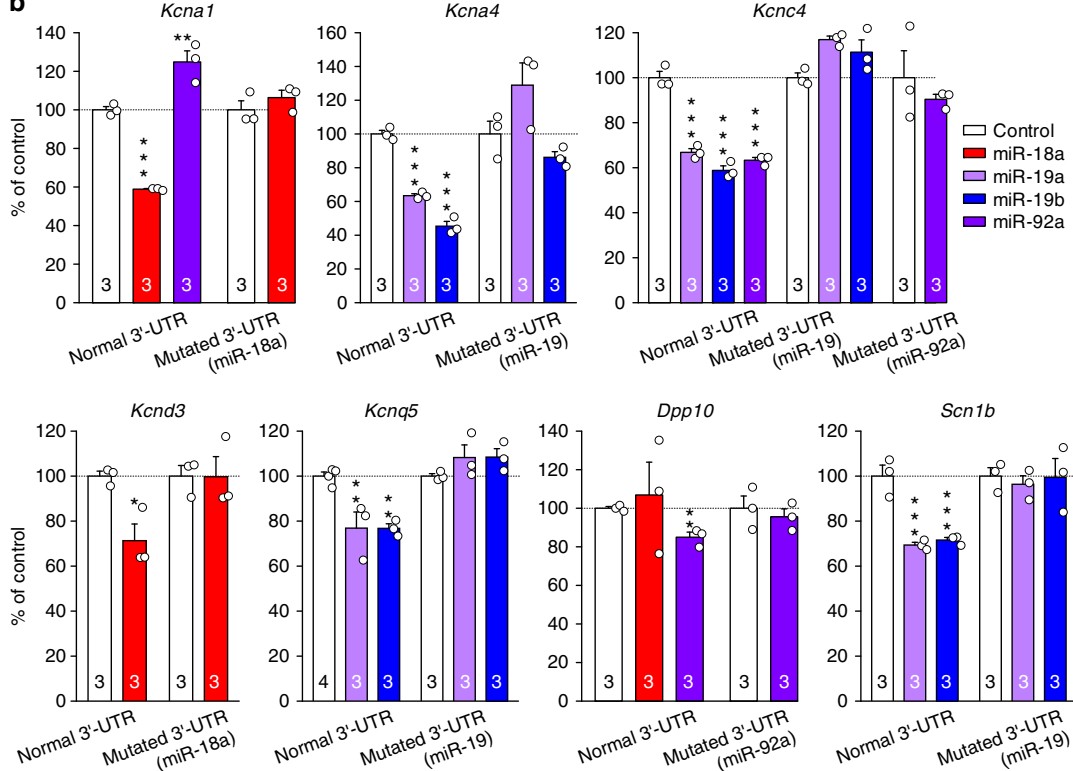

**Figure 4 | miR-17-92 cluster members differentially target the 3′-UTR sequences of voltage-gated potassium channels and modulatory subunits.**
(**a**) List of voltage-gated potassium channel subunits predicted as miR-17-92 target genes; these subunits were reportedly downregulated in a previous microarray study of SNL[20]. Parenthesized targeting miRNAs represent predicted miRNAs not validated by a luciferase assay. (**b**) Activities of luciferase with voltage-gated potassium channel subunit 3′-UTRs or mutated 3′-UTRs in HEK293 cells co-transfected with control or miR-17-92 cluster member-expressing plasmid vector. Numbers of cell cultures are indicated at the base of each bar. Error bars are s.e.m. *$P < 0.05$, **$P < 0.01$ and ***$P < 0.001$ (*Kcna1*, $P < 0.001$ for miR-18a and $P = 0.004$ for miR-92a; *Kcna4*, $P < 0.001$ for miR-19a and $P < 0.001$ for miR-19b; *Kcnc4*, $P < 0.001$ for miR-19a, $P < 0.001$ for miR-19b and $P < 0.001$ for miR-92a; *Kcnd3*, $P = 0.021$ for miR-18a; *Kcnq5*, $P = 0.008$ for miR-19a and $P = 0.008$ for miR-19b; *Dpp10*, $P = 0.006$ for miR-92a; *Scn1b*, $P < 0.001$ for miR-19a and $P < 0.001$ for miR-19b) compared with control plasmid, unpaired *t*-test or Dunnett's test.

SNL significantly reduced it (Supplementary Fig. 10). The slight reduction brought about by miR-17-92 cluster can likely be explained by its predominant targeting of the $K_V7.5$ channel among non-A-type channels in small DRG neurons. The $V_{half}$ and $k$ of total, A-type and non-A-type potassium currents were unaffected by miR-17-92 (Supplementary Fig. 11a), indicating that miR-17-92 did not affect the voltage-dependent activation of potassium channels in small DRG neurons. In medium/large DRG neurons of cell sizes $>600\,\mu m^2$, miR-17-92 did not significantly reduce total, non-A-type or A-type potassium currents (Fig. 6b,d). $V_{half}$ and $k$ in medium/large DRG neurons were also unaffected by miR-17-92 (Supplementary Fig. 11b). These results indicate that miR-17-92 has a critical role in the

function of voltage-gated potassium channels, particularly those mediating A-type currents.

**Blockade of miR-17-92 members rescues potassium currents.** To address the causal involvement of miR-17-92 in reduced potassium currents associated with neuropathic pain, the protein expressions of pore-forming potassium channel α subunits targeted by miR-17-92 were examined in the L5 DRG 28 days after SNL. L5 DRGs were obtained 28 days after SNL from rats injected with mixture of AAV vectors expressing antisense RNAs against miR-18a, miR-19a, miR-19b and miR-92a 7 days after SNL. Blockade of these miR-17-92 members significantly restored the

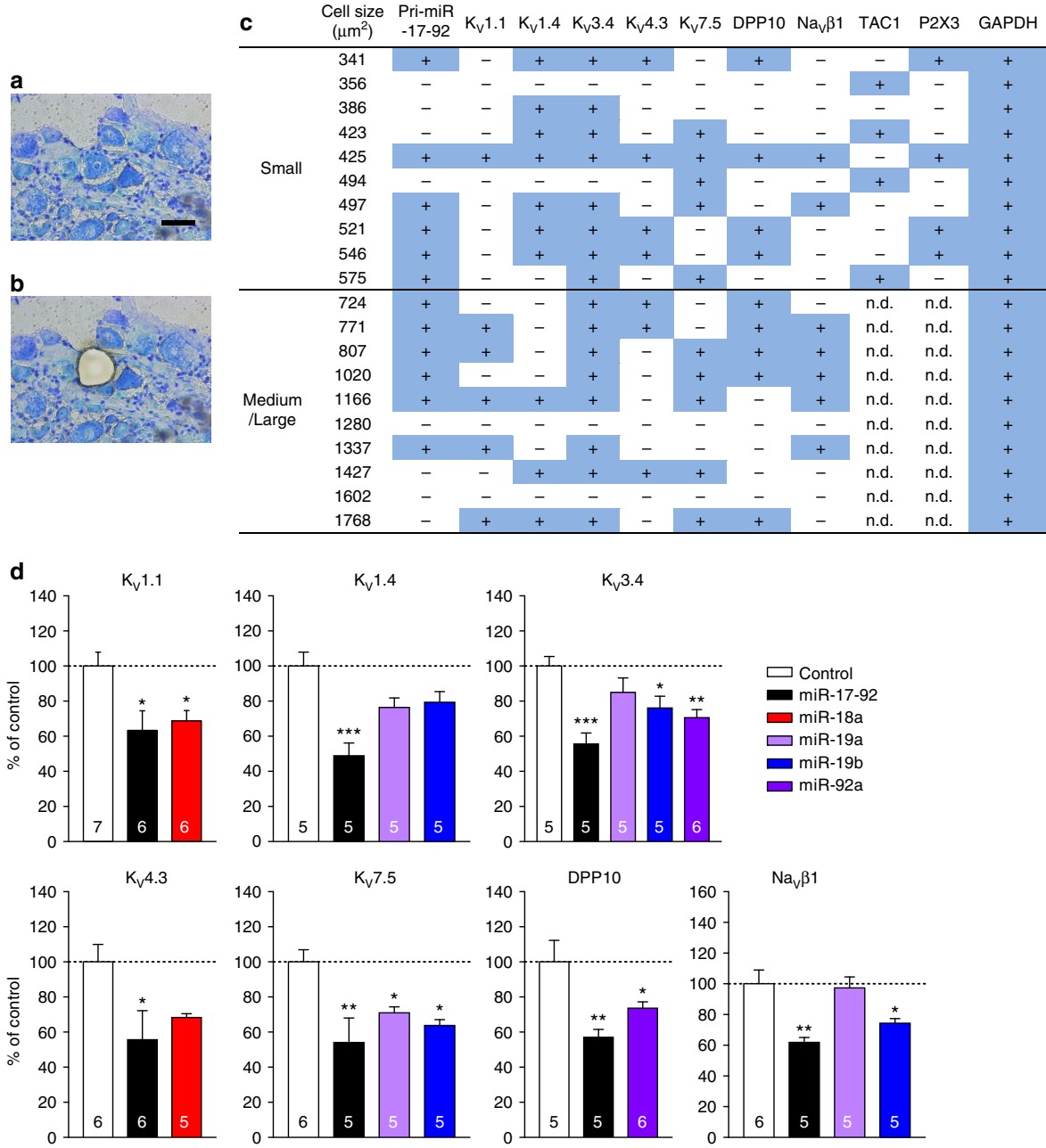

**Figure 5 | miR-17-92 cluster members differentially inhibit expression of voltage-gated potassium channel subunits *in vivo*.** (**a,b**) Representative images of L5 DRG stained with toluidine blue before (**a**) and after (**b**) laser-captured microdissection. Scale bar, 50 μm. (**c**) Expression profile of pri-miR-17-92 and voltage-gated potassium channel subunits in a single DRG neuron of indicated cell size; DRG neurons $<600\,\mu m^2$ and $>600\,\mu m^2$ were considered to be small and medium/large neurons, respectively. n.d.; not determined. (**d**) Expression of voltage-gated potassium channel subunit mRNAs examined in the L5 DRG 7 days after control or miR-17-92 AAV vector administration. Numbers of samples are indicated at the base of each bar. Error bars are s.e.m. $*P<0.05$, $**P<0.01$ and $***P<0.001$ ($K_V1.1$, $P=0.014$ for miR-17-92 and $P=0.034$ for miR-18a; $K_V1.4$, $P<0.001$ for miR-17-92; $K_V3.4$, $P<0.001$ for miR-17-92, $P=0.047$ for miR-19b and $P=0.009$ for miR-92a; $K_V4.3$, $P=0.031$ for miR-17-92; $K_V7.5$, $P=0.002$ for miR-17-92, $P=0.048$ for miR-19a and $P=0.013$ for miR-19b; DPP10, $P=0.003$ for miR-17-92 and $P=0.043$ for miR-92a; $Na_V\beta1$, $P=0.002$ for miR-17-92 and $P=0.030$ for miR-19b), Dunnett's test.

channel protein expressions (Fig. 7a,b). In line with the protein expression changes, injection of the AAV vector mixture 7 days before SNL significantly blocked the reduction in total, A-type and non-A-type potassium currents at day 7 following SNL (Fig. 7c,d).

**Potassium channel modulators suppress mechanical allodynia.** As miR-17-92 cluster suppressed the activity of multiple voltage-

gated potassium channels, we examined the effects of available potassium channel modulators on neuropathic pain. Seven days after SNL, systemic administration of flupirtine, a Kv7 potassium channel activator, partially attenuated mechanical allodynia (Fig. 8a), consistent with a previous report that flupirtine compensated for potassium channel downregulation[30]. Furthermore, NS5806, another potassium channel modulator that increases potassium current through the Kv4.3 channel, also attenuated mechanical allodynia to a similar extent (Fig. 8a). Intriguingly,

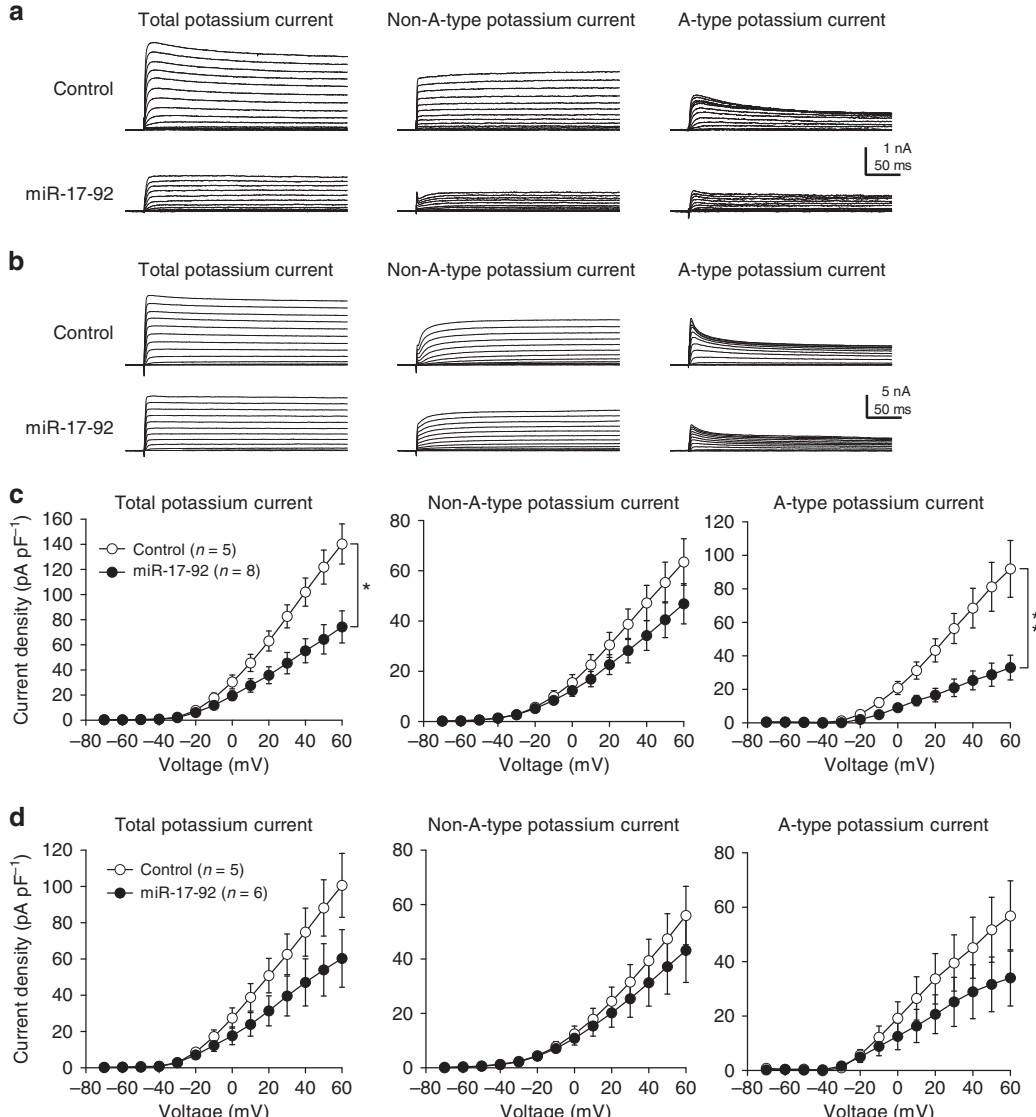

**Figure 6 | miR-17-92 reduces potassium currents in DRG neurons.** Potassium currents were recorded in acutely-dissociated DRG neurons 7 days after control or miR-17-92 AAV vector injection. Only EGFP-fluorescent DRG neurons were recorded. (**a,b**) Representative traces of total, non-A-type and A-type potassium currents elicited by stepwise depolarization from − 70 to 60 mV at holding potential of − 80 mV in small (**a**) and medium/large (**b**) DRG neurons. (**c,d**) Current density of each potassium current component plotted against voltage in small (**c**) and medium/large (**d**) DRG neurons obtained from three rats. Error bars are s.e.m. *$P = 0.016$ and **$P = 0.003$, two-way repeated-measures ANOVA.

co-administration of flupirtine and NS5806 showed more potent suppression of mechanical allodynia than each potassium channel modulator alone (Fig. 8a), suggesting the importance of concurrent enhancement of multiple potassium channels. Then, to address the correlation of voltage-gated potassium channels in mechanical allodynia mediated by miR-17-92, we administered these potassium channel modulators to rats overexpressing miR-17-92. Both flupirtine and NS5806 were partially effective (Fig. 8b), as was the case in SNL-induced allodynia. Furthermore, combination therapy with both drugs showed more potent alleviation of mechanical allodynia induced by miR-17-92 overexpression (Fig. 8b).

## Discussion

We have shown the combinatorial impact of miR-17-92 cluster members on mechanical allodynia through concurrent regulation of many functionally related voltage-gated potassium channel subunits. Furthermore, bioinformatic analysis indicated that many

genes predicted as targets of miR-17-92 were downregulated by nerve injury and were associated with neurological disease. Thus, these genes other than potassium channel subunits may also be involved in the miR-17-92-mediated mechanical allodynia. On the other hand, miR-17-92 is reported to enhance axonal growth in embryonic cortical neurons[31], which is consistently enhanced in DRG neurons affected by nerve injury. Thus, miR-17-92 may also orchestrate nerve regeneration following nerve injury. Furthermore, miR-17-92 dysregulation is reportedly observed in neurological diseases including autism spectrum disorder[3–5,32–34]. Taken together, miR-17-92 as a cluster can effectively and cooperatively regulate many aspects of physiological and/or pathological processes. Therefore, thorough investigation of the influence of multifunctional miR-17-92 cluster on neural function will likely advance our understanding of the pathophysiology of diverse neurological disorders as well as neuropathic pain.

miR-17-92 cluster members concurrently suppressed multiple voltage-gated potassium channel α subunits and modulatory

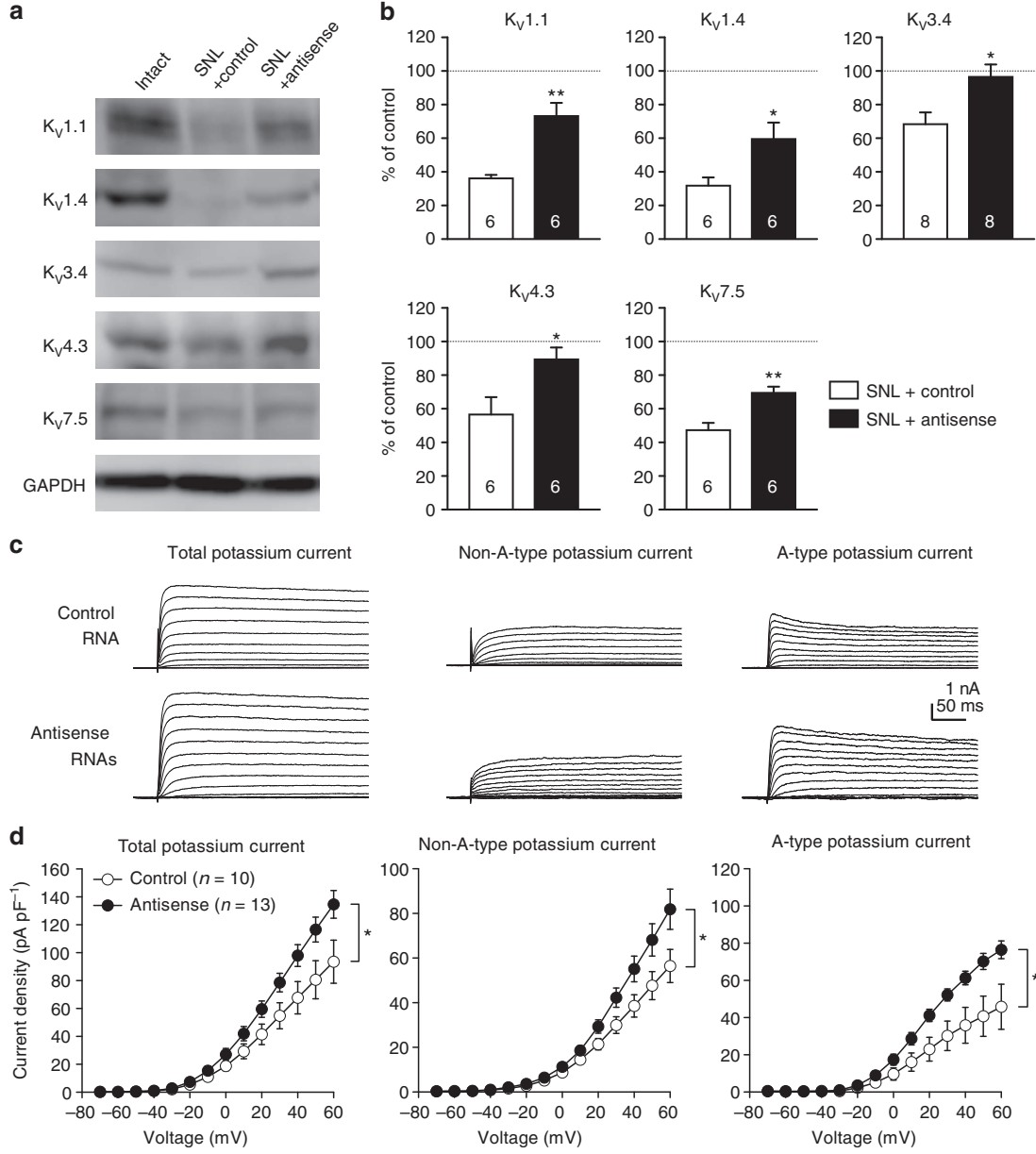

**Figure 7 | miR-17-92 inhibition rescues potassium channel expressions and potassium currents in DRG neurons in the neuropathic pain state.**
(**a**) Representative images of immunoblotting for voltage-gated potassium channel α subunits in the L5 DRG obtained from SNL rats at day 28. Full blots can be found in the Supplementary Fig. 12. AAV vectors expressing either a control AAV vector or mixture of AAV vectors encoding TuD antisense RNAs against miR-18a, miR-19a, miR-19b and miR-92a were administered 7 days after SNL. GAPDH was used as a loading standard. (**b**) Expression levels of voltage-gated potassium channel protein, as a percentage of expression level in intact L5 DRG. Numbers of samples are indicated at the base of each bar. *$P < 0.05$ and **$P < 0.01$ ($P = 0.004$ for $K_V1.1$; $P = 0.038$ for $K_V1.4$; $P = 0.015$ for $K_V3.4$; $P = 0.028$ for $K_V4.3$; $P = 0.003$ for $K_V7.5$), Welch's test. (**c,d**) Potassium currents were recorded in acutely-dissociated DRG neurons 7 days after SNL. Control AAV vector or AAV vector mixture was injected 7 days before SNL. Only EGFP-fluorescent DRG neurons were recorded. (**c**) Representative traces of total, non-A-type and A-type potassium currents elicited by stepwise depolarization from $-70$ to $60\,mV$ at a holding potential of $-80\,mV$ in small DRG neurons. (**d**) Current density of each potassium current component plotted against voltage in small DRG neurons obtained from three rats. Error bars are s.e.m. *$P < 0.05$ ($P = 0.045$ for total current; $P = 0.046$ for non-A-type current; $P = 0.015$ for A-type current), two-way repeated-measures ANOVA.

subunits in DRG neurons. Expression of these subunits detected using RT-PCR, immunohistochemistry or microarray analysis were substantially decreased in DRG neurons after nerve injury. $K_V1.1$, $K_V1.4$, $K_V3.4$ and $K_V4.3$ are reportedly downregulated in small and medium/large neurons after nerve injury[21–25]. In this study, we detected co-expression of multiple potassium genes and miR-17-92 cluster in single cells. The cell profile expressing the potassium channel subunits was comparable with previous histological studies: $K_V1.1$ was preferentially detected in

medium/large DRG neurons[24,35] and $K_V3.4$ was broadly expressed in various sized DRG neurons[22], while $K_V1.4$, $K_V4.3$, $K_V7.5$ and DPP10 were preferentially detected in small DRG neurons[22–24,36,37]. Notably, DPP10 expression was detected only in small DRG neurons that expressed $K_V4.3$, exactly consistent with a previous report[38]. In contrast, $Na_V\beta1$ distribution, which was mainly detected in medium/large DRG neurons in this study, is poorly understood, although $Na_V\beta1$ null mutant mice exhibited hyperexcitability of small DRG neurons[27]. Given that miR-17-92

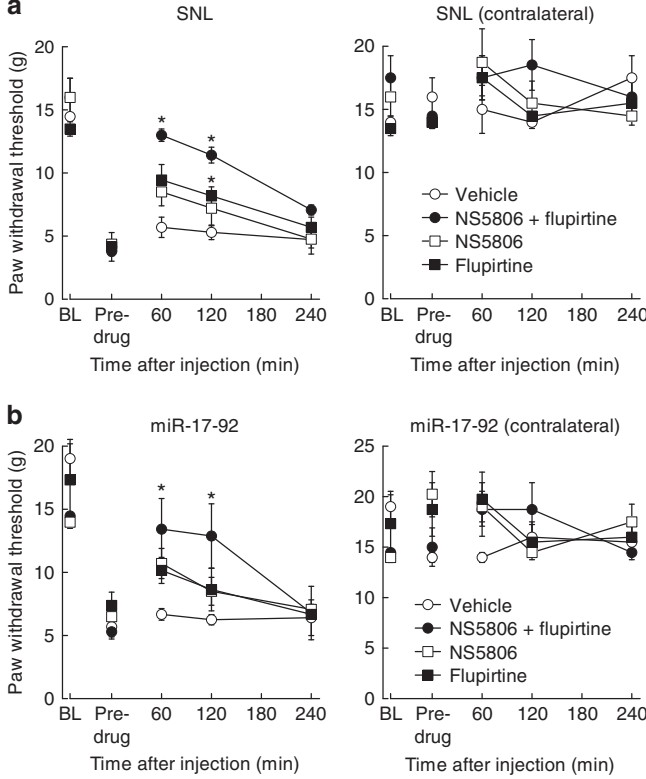

**Figure 8 | Combined administration of potassium channel modulators potently alleviated mechanical allodynia induced by miR-17-92 or SNL.** Paw withdrawal responses to mechanical stimuli were evaluated on the SNL (**a**) and AAV-injected (**b**) sides and the contralateral sides ($n = 4$). NS-5806, flupirtine or both were administered intraperitoneally 7 days after SNL; the same volume of vehicle was injected as a control. Rats were examined before SNL or AAV injection (BL), and before and after drug administration. Error bars are s.e.m. $*P < 0.05$ compared with vehicle, Steel test.

cluster members targeted 3′-UTR sequences to repress gene expression, miR-17-92 may directly decrease multiple potassium channel expression in a cell-autonomous fashion. Additionally, euchromatic histone-lysine *N*-methyltransferase-2 (G9a) mediates downregulation of most potassium channel α subunits in the DRG after nerve injury[12]. Although potassium channels modulated by G9a partially overlap with the miR-17-92 target channels, miR-17-92 and G9a modulate potassium channel expressions at distinct steps of gene expression; miR-17-92 blocks translational steps, leading to mRNA degradation, whereas G9a inhibits transcription through histone methylation. Thus, expression of potassium channels may be modulated in the injured DRG neurons through both transcriptional and post-transcriptional regulations.

miR-17-92 was particularly involved in the modulation of the fast-inactivating A-type potassium current. It is known that six $K_V$ subtypes ($K_V1.4$, $K_V3.3$, $K_V3.4$ and $K_V4.1-4.3$), three of which ($K_V1.4$, $K_V3.4$ and $K_V4.3$) are miR-17-92 targets, mediate fast-inactivating A-type currents[8]. However, $K_V4.2$ is reportedly expressed at very low levels in the DRG[21,39], further indicating the importance of miR-17-92 for fast-inactivating voltage-gated potassium channels in DRG neurons. In addition, DPP10 and $Na_V\beta1$, positive modulators of Kv4 channels, were also downregulated by miR-17-92. Dipeptidyl peptidase-like proteins (DPP6 and DPP10) are necessary for Kv4 to fully constitute the A-type current[28]. The transient outward potassium current is reduced in $Na_V\beta1$ null mutant mice with hyperexcitable DRG

neurons[27], although the reported influences of $Na_V\beta1$ on Kv4 are contradictory[28]. In line with these results, miR-17-92 significantly reduced A-type potassium currents in small, putative nociceptive DRG neurons, consistent with the reduced current in injured DRG neurons observed in this study and previous reports[12,40]. In contrast, non-A-type potassium currents were not significantly affected by miR-17-92 in small DRG neurons, although antisense RNAs restored the non-A-type potassium currents in the neuropathic pain condition. Among the voltage-gated potassium channels targeted by miR-17-92, $K_V1.1$ and $K_V7.5$ are non-A-type delayed rectifiers, but we only detected $K_V7.5$ in small DRG neurons. Similarly, medium/large DRG neurons did not show significant reduction in total, A-type and non-A-type currents after nerve injury. Consistent with this, other investigators have reported slight reduction in potassium currents in injured medium/large DRG neurons[40]. In fact, among the miR-17-92 targets only $K_V3.4$ and $K_V1.1$, which mediate A-type and non-A-type currents respectively, are expressed in medium/large DRG neurons (Fig. 5c)[22,24,36]. Overall, miR-17-92 appears particularly critical for the modulation of fast-inactivating A-type current in the nociceptive DRG neurons.

The potassium channels affected by miR-17-92 are reportedly implicated in mechanical sensation, but not heat hyperalgesia, consistent with the effects of miR-17-92 on mechanical allodynia that we observed. Chien *et al.*[22] reported that $K_V3.4$ or $K_V4.3$ knockdown with antisense oligodeoxynucleotide induced mechanical allodynia, but not thermal hyperalgesia. $K_V1.1$ is reportedly a mechanosensitive channel, inhibition of which causes mechanical allodynia, but not thermal hyperalgesia[29]. Mechanical allodynia, but not heat hyperalgesia, was reported to be mediated by small non-peptidergic C fibres[41]. Consistent with this, miR-17-92 was colocalized with $K_V3.4$ and $K_V4.3$ in non-peptidergic DRG neurons. On the other hand, ectopic discharge in large myelinated Aβ afferents is considered a source of neuropathic pain[42], while $Na_V1.8$-expressing neurons (putative small DRG neurons) were previously shown not to be essential for neuropathic pain[43]. The involvement of miR-17-92 in cold allodynia, another frequent symptom in neuropathic pain, remains unknown. Cold temperatures strongly and preferentially inhibit A-type currents but have fewer inhibitory effects on tetrodotoxin-resistant $Na^+$ channels and non-inactivating $K^+$ currents in small DRG neurons[44], suggesting a contribution of A-type $K^+$ currents to cold pain. Therefore, miR-17-92 may also have a modulatory role in nociceptive cold sensation.

Despite concerns about adverse effects on the cardiovascular and nervous systems, potassium channels are attractive therapeutic targets for neuropathic pain[7]. Flupirtine (a Kv7 activator) suppressed mechanical allodynia in the SNL model, findings consistent with other investigators[30], and a report of the successful use of flupirtine in a patient with refractory neuropathic pain due to small fibre neuropathy[45]. Importantly, in our study combined application of flupirtine and NS-5806 (a Kv4.3 activator) induced substantial relief of mechanical allodynia compared with the analgesic effect of each modulator alone. Given the downregulation of diverse potassium channel subunits in neuropathic pain, concurrent modulation of various relevant potassium channel subunits may provide superior therapeutic efficacy with fewer adverse effects. In this context, therapeutic manipulation of miR-17-92 cluster would be advantageous, as its components miR-18a, miR-19a/b and miR-92a collectively modulate multiple potassium channel α subunits and auxiliary subunits in DRG neurons. In this study, miR-17-92-mediated mechanical allodynia was alleviated by potassium channel modulators, suggesting miR-17-92 as a therapeutic target for potassium channel modulation.

In conclusion, we have shown the orchestrated effects of miR-17-92 cluster miRNAs on multiple voltage-gated potassium channels, especially those mediating A-type currents, in DRG neurons after nerve injury. Combined application of potassium channel-activating drugs *in vivo* exerted potent relief of mechanical allodynia. These findings underline the growing importance of comprehensive analysis of cluster miRNAs in the nervous system, and may provide the opportunity to develop a novel analgesic strategy based on concurrent regulation of multiple functionally related proteins.

## Methods

**Animal models.** Conduct of all experimental procedures was approved by Animal Experiments Ethical Review Committee of Nippon Medical School (Approval number, 27-037) and performed in accordance with the guidelines of the International Association for the Study of Pain[46]. Male Sprague—Dawley rats (5–6 weeks of age; Sankyo Labo Service Corporation, Tokyo, Japan) were used for all experiments. The animals were singly housed in a temperature and humidity-controlled vivarium with a 14-h/10-h light/dark cycle and allowed food and water *ad libitum*. All surgery was performed on rats under general anaesthesia induced by intraperitoneal sodium pentobarbital ($50\,mg\,kg^{-1}$) or inhaled isoflurane (2–3%). Rats were randomized to AAV or drug injection. The L5 SNL model of neuropathic pain was performed on the left side[47]. Briefly, the L5 nerve was exposed and tightly ligated with 4-0 silk thread in two regions separated by about 1 mm. To produce a spared nerve injury model of neuropathic pain, the left tibial and common peroneal nerves were tightly ligated with 4-0 silk thread and cut distal to the ligation to remove 2–3 mm of the nerve[48]. The right side was left intact as a control. As an inflammatory pain model, CFA solution ($100\,\mu l$; Sigma-Aldrich Japan, Tokyo, Japan) was injected into the left plantar skin of the hind paw innervated by L5 DRG neurons, using a 1 ml syringe with a 26-gauge needle. To assess spontaneous pain, a combined neuropathic and inflammatory pain model was developed by simultaneously performing SNL and injecting the left paw with $50\,\mu l$ of CFA[18]. The right side was left intact as a control.

The potassium channel modulators flupirtine (Tocris Bioscience, Bristol, UK) and NS5806 (Sigma-Aldrich) were administered intraperitoneally at doses of $20\,mg\,kg^{-1}$ and $5\,mg\,kg^{-1}$, respectively, 7 days after SNL or AAV vector injection. Flupirtine and NS5806 were dissolved in dimethyl sulfoxide and diluted with physiological saline at concentrations of $4\,mg\,ml^{-1}$ and $1\,mg\,ml^{-1}$, respectively, before use. Vehicle was used as a control.

**Behavioural tests.** Paw withdrawal responses to mechanical stimuli were measured using a set of von Frey filaments (Muromachi Kikai, Tokyo, Japan). Each rat was placed on a metallic mesh floor covered with a plastic box, and a von Frey monofilament was applied from under the mesh floor to the plantar surface of the hind paw. The smallest force inducing withdrawal of the stimulated paw at least three times in five trials was referred to as the paw withdrawal threshold. Rats that had undergone SNL but that did not show reduction in the paw withdrawal threshold ($>7\,g$) 7 days later were excluded from the analysis. The Plantar Test (Ugo Basile, Varese, Italy) was used to examine thermal hyperalgesia. After von Frey test, each rat was placed on a glass plate with a radiant heat generator underneath. The latency of paw withdrawal from the heat stimulus was measured on two occasions 5 min apart, and the average was referred to as the paw withdrawal latency. Spontaneous paw liftings were counted for 10 min 7 days after SNL + CFA treatment. Most behavioural tests were performed by an investigator blind to experimental conditions.

The open field test was performed 6 days after the AAV injection. The open field chamber was $100 \times 100\,cm^2$ (length × width) (O'Hara & Co., Ltd., Tokyo, Japan). The field was illuminated at 40 lux. Each rat was allowed to explore the novel open field for 10 min. Total distance travelled was determined using Image J OFC (O'Hara & Co., Ltd.) comprising modified software based on Image J.

The rotarod test was performed using an accelerating rotarod apparatus (O'Hara & Co., Ltd.). Rats were acclimatized to the apparatus on day 6 after the open field test. On the next day, the rats were put onto the accelerating rod from 3 to 30 r.p.m. over a period of 300 s. Five trials at intervals of $>30\,min$ were performed for each rat and the mean time to fall (in s) was obtained.

**Quantitative PCR.** Reagents and kits were provided by Life Technologies (Carlsbad, CA) unless otherwise stated. All procedures were performed according to the manufacturers' protocols. Total RNA was extracted from the L4 and L5 DRGs and L5 dorsal spinal cord using RNAiso plus (Takara Bio, Shiga, Japan) or *miRVana* PARIS kit according to the manufacturers' protocols. For miRNA quantification, total RNA (10 ng) was reverse-transcribed with a stem-loop primer specific for each mature miRNA using a TaqMan MicroRNA Reverse Transcription Kit. PCR mixture was prepared using TaqMan Universal PCR Master Mix and premixed TaqMan probe and primer pairs specific for each miRNA included in the TaqMan MicroRNA Assays (Supplementary Table 2). For quantification of mRNAs and pri-miRNAs, total RNA (500 ng) was reverse-transcribed using iScript

select cDNA Synthesis Kit (Bio-Rad Laboratories, Hercules, CA) with a random primer. PCR amplification was performed with TaqMan Gene Expression Master Mix using a premix of gene-specific TaqMan probe and primer pairs (Supplementary Table 2). In the case of *Tac1* amplification, the forward primer (5′-CGCAATGCAGAACTACGAAAGA-3′), reverse primer (5′-CGCGGACACA-GATGGAGAT-3′) and probe (5′-CGTAAA-TAAACCCTGTAACGCACTATCTAT-3′) were used. The amplification efficiency per single PCR cycle was obtained by assaying serially-diluted samples (four points at 1:5 dilution) and the relative expression was calculated.

**RT-PCR of microdissection samples.** A single cell was obtained by laser capture microdissection. The L5 DRG was excised and then rapidly frozen in OCT compound (Sakura Finetek, Tokyo, Japan) using dry ice/acetone. The DRG was sectioned ($20\,\mu m$) using a cryostat (Leica Microsystems Wetzlar, Germany) and placed on an RNase-free glass slide with PEN-membrane (Leica Microsystems). The section was incubated in RNA later solution (Life Technologies) for 2 min at room temperature. After wash with RNase-free water for 1 min on ice, the section was stained with toluidine blue (0.05%, pH 4.1) for 15 s at $4\,°C$ to visualize the DRG neurons. After washing with RNase-free water for 1 min on ice, the section was dried with a hair drier. A single DRG neuron was obtained using a laser microdissection system (LMD7000; Leica Microsystems).

The single DRG neuron was processed with Single Cell-to-CT kit according to the manufacturer's protocol (Life Technologies). Briefly, the single DRG neuron was lysed and the DNA was degraded with DNase I. After RT reaction of the lysed sample, preamplification was performed using cocktailed TaqMan Gene Expression Assays including those for primary miR-17-92 and voltage-gated potassium channels. PCR amplification of the diluted preamplified product was conducted as described in the Quantitative PCR section above. More than ten-fold signal intensity compared with negative control was considered as positively detected.

**Plasmids.** To express miRNAs as a cluster or individually, sequences encoding the miRNAs were amplified from rat genome using primer pairs with EcoRI restriction sites attached at the 5′ ends (Supplementary Table 3). The amplified sequences were subcloned into the EcoRI site of the AAV vector plasmid pAAV-EGFP[49], which contains the CAG promoter upstream of the transgene and the EGFP gene driven by the B19 promoter. pAAV-EGFP[50], which contains the EGFP gene driven by the CAG promoter, was used as a control. TuD RNA was used to inhibit miRNA function[17]. A TuD sequence less homologous to known miRNAs was used as a negative control (Takara Bio). Clone IDs of TuD were as follows: NC000001 (negative control), RH000611 (miR-17), RH000323 (miR-18a), RH000643 (miR-19a), RH000352 (miR-19b), RH000277 (miR-20a) and RH000184 (miR-92a). Mouse U6 promoter and anti-miRNA TuD sequence was cut with ClaI from a pBAsi-Pur plasmid vector encoding TuD RNA against individual miRNA (Takara Bio). The ClaI fragment was subcloned into the ClaI site of pBluescript II whose BamHI site was mutated to Acc65I. This plasmid vector was further digested with Acc65I and the resultant fragment was subcloned into Acc65I site of pAAV-EGFP plasmid.

Plasmid vectors encoding firefly luciferase followed by the 3′-UTR of voltage-gated potassium channels were constructed. Each 3′-UTR sequence was amplified from rat DRG-derived cDNA using primer pairs with SpeI or HindIII restriction sites attached at the 5′ ends (Supplementary Table 3). The 3′-UTR sequences were subcloned into the SpeI or HindIII site (after the luciferase gene) of pMIR-REPORTER. The putative seed sequences within the 3′-UTR sequences were mutated using QuikChange II XL Site-Directed Mutagenesis Kit (Agilent Technologies, Santa Clara, CA).

**AAV vector production.** Serotype 6 AAV vectors were produced by adenovirus-free triple transfection with AAV vector, AAV packaging (pRepCap 6as; kindly provided by Dr DW Russell)[51] and helper (pHelper; Agilent Technologies) plasmids at a ratio of 1:1:1. For miRNA-expressing AAV vector and its control, plasmids were co-transfected into HEK293 cells (ATCC, Manassas, VA) using calcium phosphate precipitation as previously described[52]. Six hours after transfection, the culture medium was refreshed and cells were cultured for 3 days at $37\,°C$ in a humidified atmosphere of air and 5% $CO_2$. Cells were suspended in PBS (phosphate buffered saline) and freeze-thawed three times. Cell debris was pelleted by centrifugation at 6,000 r.p.m. for 30 min at $4\,°C$, and AAV vectors were purified by ammonium sulfate precipitation and iodixanol continuous gradient centrifugation. AAV vectors expressing TuD antisense RNAs or its control were produced with ultracentrifugation-free chromatography-mediated purification, as previously described[52,53]. Plasmids were co-transfected into HEK293 cells using polyethylenimine and were cultured for 5 days. The culture medium was cleared by activated charcoal followed by centrifugation at 3,000 r.p.m. for 20 min at $4\,°C$. AAV vector was concentrated by ultrafiltration with a hollow fibre 750 kDa filter and then by CsCl density gradient centrifugation. After purification by dialysis, AAV vector was further concentrated with Amicon Ultra-4 30 K (Merck Millipore, Darmstadt, Germany). The titre of the AAV vector was determined by qPCR. For use, each AAV vector was diluted with PBS to $\sim 5 \times 10^{13}$ vector genomes (vg) $ml^{-1}$.

To inject an AAV vector, rats were laid prone under deep anaesthesia. Then, paraspinal muscles were separated from the vertebrae and a small part of vertebrae overlying the L5 DRG was removed to expose the ganglion. AAV vector (5 µl) was slowly (>5 min) injected into the left L5 DRG using a microsyringe with a curved-tip 27-gauge needle. Rats exhibiting motor disturbance were excluded from analysis. This procedure specifically introduces transgene into L5 DRG neurons of all cell sizes, as previously described[6]. *In vivo* transduction into the DRG neurons was confirmed for all AAV vectors by immunofluorescence for EGFP. A lack of apparent motor dysfunction assessed by the open field and rotarod tests (Supplementary Fig. 6b,c) and no change in thermal withdrawal latency (Supplementary Fig. 4) by miR-17-92 overexpression or inhibition indicated that the AAV injection itself and miR-17-92 in the DRG neurons did not obviously affect motor function.

**Immunofluorescence.** Rats were perfused transcardially with PBS (pH 7.4) followed by fresh 4% paraformaldehyde in PBS. DRGs were removed and post-fixed in the same fixative overnight, and then immersed in 20% sucrose in PBS. On the next day, DRGs were rapidly frozen in dry ice/acetone and sectioned (10 µm) using a cryostat (Leica Microsystems). The DRG sections were pre-incubated in PBS containing 5% normal donkey serum and 0.2% Triton X-100 for 30 min, and then incubated with a rabbit anti-green fluorescent protein antibody (1:1000; A11122, Life Technologies) at 4 °C overnight. After washing with PBS, sections were incubated with a secondary antibody labelled with Alexa Fluor 488 at room temperature for 1 h. Images were captured using a high-resolution digital camera equipped with a computer (Olympus, Tokyo, Japan). To measure the size of DRG neurons, two DRG sections (minimum separation, 60 µm) obtained from individual rats were analysed using ImageJ software (National Institutes of Health, Bethesda, MD).

**Luciferase assay.** Activities of firefly and *Renilla* luciferases were measured using the Dual-Glo Luciferase Assay System (Promega, Fitchburg, WI). HEK293 cells (System Biosciences, Palo Alto, CA) were cultured in DMEM medium supplemented with 10% heat-inactivated fetal bovine serum and antibiotics (penicillin and streptomycin). The cells were seeded onto a white 96-well plate ($2 \times 10^4$ cells per well). To assess TuD antisense RNA efficiency, HEK293 cells were transfected with pAAV-EGFP plasmids, prior to transfection for luciferase assays. The next day, cells were co-transfected with pMIR vector with 3′-UTR, pGL4.74[hRluc/TK] vector (Promega) and pAAV vector using Lipofectamine2000. Two days after transfection, Dual-Glo luciferase reagent was added to each well and firefly luminescence was measured using Wallac 1420 ARVO$_{MX}$ (PerkinElmer, Waltham, MA). Dual-Glo Stop & Glo reagent was then added to each well and *Renilla* luminescence was measured. Firefly luminescence was divided by *Renilla* luminescence and luminescence of control pMIR vector for normalization.

**Microarray.** Total RNA was isolated from L5 DRG using RNAiso Plus. Cy3-labelled cRNA was prepared from total RNA (200 ng) using the Low Input Quick Amp Labeling Kit according to the manufacturer's protocol (Agilent Technologies). After purification, cRNA was hybridized overnight to a rat microarray slide (SurePrint G3 Rat GE 8 × 60 K; Agilent Technologies) at 10 r.p.m. and at 65 °C. Fluorescent images of the microarray slide were scanned using a DNA Microarray Scanner (Agilent Technologies). The fluorescent intensity of each spot was quantified using Feature Extraction software (Agilent Technologies). Data were analysed using GeneSpring GX software (Agilent Technologies).

**Electrophysiology.** L5 DRGs were removed from rats 7 days after SNL or AAV vector injection and immersed in Ham's F12 nutrient mixture. DRGs were cut into small pieces and incubated in PBS containing collagenase A (5 mg ml$^{-1}$; Roche Diagnostics, Basel Switzerland) and dispase II (1 mg ml$^{-1}$; Roche Diagnostics) for 30 min at 37 °C. The solution was replaced with F12 medium and the DRGs were dissociated by gentle pipetting. After two washes with F12 medium, the cell suspension was placed onto a glass coverslip coated with poly-D-lysine and laminin (BD Biosciences, Franklin Lakes NJ, CA). After incubation at 37 °C in a humidified atmosphere of air and 5% CO$_2$ for 1–2 h, electrophysiological recording was performed.

The coverslip was set on a submersion-type recording chamber perfused with extracellular solution (150 mM choline chloride, 5 mM KCl, 2 mM CaCl$_2$, 1 mM MgCl$_2$, 10 mM HEPES, 1 mM CdCl$_2$ and 10 mM D-glucose, pH 7.4) bubbled with oxygen. Borosilicate glass-patch electrodes (World Precision Instruments, Sarasota, FL) with a resistance of 3–5 MΩ when filled with an internal solution of 150 mM potassium methanesulphonate, 5 mM KCl, 0.5 mM EGTA, 10 mM HEPES, 5 mM Mg-ATP and 0.4 mM Na-GTP (pH 7.4) were used for whole-cell recordings of DRG neurons. Membrane currents in the whole-cell configuration were acquired with Axon 700B Multiclamp amplifier and pClamp acquisition software (Molecular Devices, Sunnyvale, CA). In DRG neurons obtained from rats injected with AAV, only EGFP-positive AAV-transfected neurons were examined. The total potassium current was recorded during a series of depolarizing voltages from − 80 to 60 mV (400-ms pulse duration) in 10 mV increments at 6 s intervals. Leak currents were subtracted using the online P/8 protocol. Then, after blocking the A-type potassium channel by bath application of 5 mM DAP[12] for ∼5 min, the non-A-

type potassium current was recorded with the same depolarizing protocol. The DAP-sensitive A-type potassium current was obtained by subtracting the non-A-type current remaining in the presence of DAP from the total potassium currents without DAP. Whole-cell current–voltage (*I–V*) curves were obtained by measurement of the peak outward current at each depolarizing potential and normalized to the cell capacitance. The values of V$_{half}$ and *k* were calculated from the Boltzmann equation after converting to conductance-voltage curves. All the electrophysiological experiments were performed by an investigator blind to experimental conditions.

**Immunoblotting.** L5 DRGs were sonicated in 10 mM Tris–HCl (pH 7.2) containing 250 mM sucrose, 10 mM HEPES, 10 mM EDTA and protease inhibitor cocktail (Roche Diagnostics). Homogenates were centrifuged at 12,000 r.p.m., at 4 °C for 20 min. Supernatants were electrophoresed on SDS-polyacrylamide gels and electroblotted onto PVDF membranes (GE Healthcare). Membranes were incubated with anti-K$_V$1.1 (1:300; K20/78, NeuroMab, Davis, CA), anti-K$_V$1.4 (1:300; K13/31, NeuroMab), anti-K$_V$3.4 (1:200; APC-019, Alomone Labs, Jerusalem, Israel), anti-K$_V$4.3 (1:100; K75/41, NeuroMab), anti-K$_V$7.5 (1:200; APC-155, Alomone Labs) or anti-GAPDH (1:1,000; 14C10, Cell Signaling Technology, Danvers, MA) antibody at 4 °C overnight and then detected with HRP-conjugated secondary antibody (1:2,000; Cell Signaling Technology) and chemiluminescence (ECL Prime Western Blotting Detection Reagents; GE Healthcare). The luminescence was detected with a C-DiGit Blot Scanner (LI-COR Biotechnology, Lincoln, NE). GAPDH was used as a loading standard. Optical densities of bands were quantified using Scion Image Beta 4.03. Full images of the blots are shown in Supplementary Fig. 12.

**Statistics.** Values are expressed as mean ± standard error. SPSS (version 18, IBM, Armonk, NY) and KyPlot (KyenceLab, Tokyo, Japan) were used for statistical analyses. Sample sizes were not statistically estimated but were adopted to minimize the number of rats used. Normality of data was assessed by the Shapiro-Wilk test. Equality of variance was assessed by Levene's test. The paired *t*-test, unpaired *t*-test and one-way factorial ANOVA followed by Dunnett's test for multiple comparisons were used for normally distributed data sets with equal variance. Welch's test was used for normally distributed data sets when equality of variance was rejected. When normality was rejected, the Mann–Whitney *U* test or the Steel test for multiple comparisons was used. Differences between groups were assessed using two-way repeated-measures ANOVA. All tests were two-tailed and *P* values <0.05 were considered statistically significant.

**Data availability.** Microarray data have been deposited in Gene Expression Omnibus (GSE98636). All other data are available on request from the authors.

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

## Acknowledgements

We thank Yasunori Mikahara, Kumi Adachi, Kumiko Takasu, Rina Takata, Kohki Shibata, Asami Sugita and Chiaki Saitoh for their technical assistance. We also thank Toshihiro Takizawa for assistance with the bioinformatic analysis and Masatoshi Nagano for guidance with open field and rotarod tests. A.S.s work was supported by a Grant-in-Aid for Scientific Research (C25462454) from the Japan Society for the Promotion of Science; H.S.s work by a Ministry of Education, Culture, Sports, Science and Technology Supported Program for the Strategic Research Foundation at Private Universities 2008–2012, Japan (S0801035).

## Author contributions

A.S. and H.S. designed and analysed the experiments and wrote the manuscript. A.S. and M.M. performed the animal experiments and microdissection. A.S. performed the qPCR, immunofluorescence, luciferase assay and immunoblotting. N.M., K.M., T.S. and T.O. produced the AAV vectors. F.S. performed and analysed the electrophysiological experiments.

## Additional information

**Competing interests:** The authors declare no competing financial interests.

