## [Peer Review File · Nature Communications]

Reviewers' comments:

Reviewer #1 (Remarks to the Author):

The authors in this study determined the role of miR-17-92 cluster in regulating K⁺ channel expression in the DRG using a rat model of neuropathic pain. They showed that L5 spinal nerve ligation upregulated all 6 members of the miR-17-92 cluster in the rat DRG. Overexpressing the miR-17-92 cluster or one of the 4 members (miR-18a, 114 miR-19a, miR-19b or miR-92a) via an AAV vector injected into the DRG induced tactile allodynia in rats. Blocking miR-18a, 114 miR-19a, miR-19b or miR-92a using antisense RNA before nerve injury reduced subsequent allodynia. Furthermore, overexpression of the miR-17-92 cluster in the DRG reduced the expression and currents of several voltage-gated K⁺ channels. In addition, they showed that the potassium channel modulators reduced allodynia induced by nerve injury or miR-17-92 overexpression. Overall, this seems to be an interesting study, and the conclusion is generally supported by the available data. This information add some new knowledge to the understanding of neuropathic pain mechanisms. The manuscript is generally well written, and the presentation is clear. Nevertheless, the authors need to address the following major points to strengthen their conclusion and increase the impact of the study.

1. The authors need to show whether blocking the miR-17-92 cluster (or miR-18a, 114 miR-19a, miR-19b or miR-92a) using antisense RNA can block nerve injury-induced downregulation of K⁺ channel expression and their currents in DRG neurons. This evidence is critical for the authors' conclusion about the direct link between miR-17-92 and K⁺ channels in injured DRG neurons. At the present form, the conclusion is largely based on indirect and correlative data using overexpression and K⁺ channel modulators. However, K⁺ channel modulators/openers can reduce neuronal excitability and pain even in the absence of altered K⁺ channel expression.

2. The information about the relationship between miR-17-92 and K⁺ channels is incomplete. Injection of an AAV vector encoding whole miR-17-92 cluster did NOT decrease the expression of ALL voltage-gated K⁺ channel subunits, as the authors claimed (line 191). Fig. 5 only listed 7 K⁺ channel subunits affected by this approach, and the target genes were largely predicted from bioinformatic analysis (Fig. S6). The authors should use an unbiased microarray or siRNA approach to show the gene profile altered by miR-17-92 overexpression (or antisense treatment in nerve injured rats) in the DRG. There are more than 70 K⁺ channel genes, and it is desirable to know whether miR-17-92 affects the expression of non-voltage-gated K⁺ channels (e.g., ATP-gated and Ca²⁺-activated) and non-K⁺ channel genes. These results are essential to strengthen the conclusion and to increase the general impact/implications of this study.

3. The authors injected the antisense RNA 7 days BEFORE spinal nerve ligation in Fig. 6. To establish the role of miR-17-92 in maintaining nerve injury-induced chronic pain (and for the therapeutic purpose), they need to show whether antisense injection can reverse nerve injury-induced allodynia and K⁺ channel downregulation.

4. The epigenetic regulator G9a has been shown to mediate almost all K⁺ channel downregulation in the DRG after nerve injury. Does G9a inhibition normalize the miR-17-92 expression in the injured DRG? The authors need to explain any potential connections between the epigenetic factors (e.g., G9a) and the miR-17-92 in reducing K⁺ channel expression in the injured DRG.

5. Data in Fig. 1 is confirmatory and has been reported in the authors' previous work. The data in Fig. 7 are correlative at best, as explained above. These two figures could be moved to supplementary materials. Also, Fig. 5 was not presented in order (directly after Fig. 2) on page 5.

Reviewer #2 (Remarks to the Author):

This is an interesting manuscript. The authors found that miR-17-92 cluster members are up-regulated in DRG neurons after spinal nerve ligation (SNL). Overexpressing of miR-17-92 cluster and some individual members such as miR-18a, miR-19a, miR-19b, or miR-92a (but not miR-17 and miR-20a) induced mechanical allodynia, but thermal hyperalgesia. Knocking down miR-17-92 cluster suppressed SNL-induced mechanical allodynia, but not thermal hyperalgesia. Several voltage-gated potassium channels KV1.1, KV1.4, Kv3.4, KV4.3 and positive modulators DPP10 and Navβ1 and their potassium currents were affected by miR-17-92 in DRG neurons. Single cell analysis showed the co-expression of miR-17-92 with these channels. These findings indicate that miRNA cluster miR-17-92 may participate in the mechanism of nerve injury-induced pain. The manuscript was well written. However, there are some concerns need to be properly addressed or discussed.

Major points:

miR-17-92 expresses all types of DRG neurons. Nevertheless, it reduced Kv currents in small, but not medium/large, DRG neurons. The behavioral study showed that mi-17-92 affected mechanical allodynia, but not thermal hyperalgesia. How does this happen? The statement "the potassium channels affected are reportedly implicated only in mechanical sensation" (page 9, line 196-198) is improper. DRG Kv1.2 overexpression or knockdown affected paw withdrawal latency to thermal stimulation (Fan L et al., Mol Pain, 2014; Zhao X et al., Nature Neurosci 2013; Li Z et al., Pain 2015). More detailed discussion should be provided.

In addition, all changes in the channel expression given are at the levels of mRNAs. The changes in the amounts of their proteins should be provided.

Minor points:

1. The accession number of Gene Expression Omnibus data used for bioinformatics analysis is missing.
2. Supplementary Table 1 is missing.
3. miR-17-92 was up-regulated in the DRG after SNL, but not after CFA-induced inflammation. Does this phenomenon occur in other neuropathic pain models or only in SNL model? Moreover, the authors showed this up-regulation in the L5 DRG after SNL. It is unknown if such up-regulation also occurs in the intact DRG (i.e. ipsilateral L4 DRG) and the ipsilateral L5 spinal cord.
4. DRG neurons are usually divided into three categories: small (< 600 μm²), medium (600-1,200 μm²) and large (> 1,200 μm²). Some medium-sized DRG neurons are c-fiber or Aδ-fiber ones.
5. AAV usually requires several (at least 2-3 weeks) weeks to become expressed in vivo. It is interesting that the authors reported an increase in miR-17-92 at 7 days after AAV injection. There is no time course in the changes of its expression and behavioral responses after AAV injection.
6. A total of 5 ul AAV was injected with a 27-gauge microsyringe. Such volume and microsyringe may damage DRG cells significantly.
7. Does the altering miR-17-92 expression affect motor function?
8. In addition to mechanical allodynia, other typical neuropathic symptoms are cold allodynia and spontaneous ongoing pain. Does the miR-17-92 affect those symptoms?
9. The single DRG neuron RT-PCR results showed co-expression of multiple potassium genes and miR-17-92 cluster. It is rough to measure the size of the neurons to classify them as C-fiber or A-fiber. Which internal control was used as the indicator of successful isolation of RNA from each neuron by LCM? miRNA in situ hybridization and immunofluorescence experiments will further to support these conclusions.
10. The method of cell culture is missing.
11. The detailed information of primers and the catalog number of qPCR kit are missing.
12. Please indicate the sequences of TuD RNA against individual miRNA or relevant catalog numbers in the Methods part.
13. Some unnecessary open circles are found in Fig 2b and 4b.

Reviewer #3 (Remarks to the Author):

Sakai et al. have demonstrated that the miR-17-92 cluster members were persistently upregulated in primary sensory neurons after nerve injury. Overexpression of miR-18a, miR-19a, miR-19b and miR-92a cluster members elicited mechanical allodynia in intact rats, while their blockade alleviated mechanical allodynia in a rat model of neuropathic pain. They also found that plausible targets for the miR-17-92 cluster included genes encoding numerous voltage-gated potassium channels and their modulatory subunits. Their unique and simple single cell analysis revealed extensive co-expression of miR-17-92 cluster and its predicted targets. Under this condition, miR-17-92 downregulated the expression of genes encoding potassium channels, and reduced outward potassium currents (especially A-type currents). In addition, combined application of potassium channel modulators synergistically alleviated mechanical allodynia induced by nerve injury or miR-17-92 overexpression, indicating miR-17-92 cluster may cooperatively regulate the function of multiple voltage-gated potassium channel subunits, which is associated with mechanical allodynia. They conclude that the orchestrated effect of miR-17-92 cluster miRNAs on multiple voltage-gated potassium channels, especially those mediating A-type currents, in DRG neurons is critical for neuropathic pain after nerve injury. They insist that combined application of potassium channel-activating drugs could exert potent relief of mechanical allodynia. Although the authors presented solid experiments containing interesting findings, there are a couple of concerns, in my opinion, that need to be addressed so as to increase the impact of this manuscript.

Specific comments:

1. The study for acute pain, instead of Day 3, is required to understand the late phase of such epigenetic modulation in the development of neuropathic pain sensation.
2. The authors have nicely done the reporter gene assay. However, instead of RT-PCR, the western blotting assay should be required to determine the influence of miRNAs on targeted mRNAs. Simple comparison analysis in the complex modulation to target mRNAs/protein synthesis by miRNAs between RT-PCR and western blotting might be informative.
3. A simple single cell analysis is unique and interesting. However, the present information of "cell heterogeneity" may not be enough to identify the cell types (ex: peptidergic or non-peptidergic cell). The authors should identify each cell type.

Responses to Reviewer #1

Comment 1: The authors need to show whether blocking the miR-17-92 cluster (or miR-18a, miR-19a, miR-19b or miR-92a) using antisense RNA can block nerve injury-induced downregulation of K⁺ channel expression and their currents in DRG neurons. This evidence is critical for the authors' conclusion about the direct link between miR-17-92 and K⁺ channels in injured DRG neurons. At the present form, the conclusion is largely based on indirect and correlative data using overexpression and K⁺ channel modulators. However, K⁺ channel modulators/openers can reduce neuronal excitability and pain even in the absence of altered K⁺ channel expression.

To address the reviewer's comment, we investigated the effects of antisense RNAs on K⁺ channel protein expressions and their currents and found that concurrent injection of AAV vectors expressing miR-18a, miR-19a, miR-19b and miR-92a blocked the protein expressions and K⁺ currents (Fig. 7; page 11, line 20 to page 12, line 5).

Comment 2: The information about the relationship between miR-17-92 and K⁺ channels is incomplete. Injection of an AAV vector encoding whole miR-17-92 cluster did NOT decrease the expression of ALL voltage-gated K⁺ channel subunits, as the authors claimed (line 191). Fig. 5 only listed 7 K⁺ channel subunits affected by this approach, and the target genes were largely predicted from bioinformatic analysis (Fig. S6). The authors should use an unbiased microarray or siRNA approach to show the gene profile altered by miR-17-92 overexpression (or antisense treatment in nerve injured rats) in the DRG. There are more than 70 K⁺ channel genes, and it is desirable to know whether miR-17-92 affects the expression of non-voltage-gated K⁺ channels (e.g., ATP-gated and Ca²⁺-activated) and non-K⁺ channel genes. These results are essential to strengthen the conclusion and to increase the general impact/implications of this study.

miR-17-92 decreased expression of all voltage-gated potassium channel subunits targeted by miR-17-92, and we therefore revised this sentence for clarity (page 9, lines 22–24). We also examined the other K⁺ channels that are not targeted by the miR-17-92 cluster, including ATP-gated (K_{ir}6.1), Ca²⁺-activated (BK_{CA}) and K_v1.2 channels. We found that these K⁺ channel expressions were not significantly changed by miR-17-92 overexpression (Supplementary Fig. 9; page 10, lines 5–7), suggesting that miR-17-92 specifically modulates its target potassium channels.

Comment 3: The authors injected the antisense RNA 7 days BEFORE spinal nerve ligation in Fig. 6. To establish the role of miR-17-92 in maintaining nerve

injury-induced chronic pain (and for the therapeutic purpose), they need to show whether antisense injection can reverse nerve injury-induced allodynia and K⁺ channel downregulation.

In accordance with the reviewer's suggestion, we injected AAV vectors expressing antisense RNAs 7 days AFTER spinal nerve ligation when the neuropathic pain had been established, and assessed the effect on the mechanical allodynia. miR-17-92 antisense RNAs also reversed the mechanical allodynia (Fig. 3b; page 6, lines 22–24) and potassium channel expressions (Fig. 7a; page 11, line 20 to page 12, line 2).

Comment 4: The epigenetic regulator G9a has been shown to mediate almost all K⁺ channel downregulation in the DRG after nerve injury. Does G9a inhibition normalize the miR-17-92 expression in the injured DRG? The authors need to explain any potential connections between the epigenetic factors (e.g., G9a) and the miR-17-92 in reducing K⁺ channel expression in the injured DRG.

To address the reviewer's comment, we added the following discussion of potential connections between G9a and miR-17-92 in K⁺ channel expression to the revised manuscript (page 14, lines 13–21): “Euchromatic histone-lysine *N*-methyltransferase-2 (G9a) mediates downregulation of most potassium channel α subunits in the DRG after nerve injury. Although potassium channels modulated by G9a partially overlap with the miR-17-92 target channels, miR-17-92 and G9a modulate potassium channel expressions at distinct steps of gene expression; miR-17-92 blocks translational steps, leading to mRNA degradation, whereas G9a inhibits transcription through DNA methylation. Thus, expression of potassium channels may be modulated in the injured DRG neurons through both transcriptional and post-transcriptional regulations.”

Comment 5: Data in Fig. 1 is confirmatory and has been reported in the authors' previous work. The data in Fig. 7 are correlative at best, as explained above. These two figures could be moved to supplementary materials. Also, Fig. 5 was not presented in order (directly after Fig. 2) on page 5.

As the reviewer indicated, our previous microarray analysis had already suggested the increases in miR-17-92 cluster members in the L5 DRG 14 days after SNL. However, the microarray results of miR-17-92 cluster members had not been validated by the more rigorous analysis of quantitative PCR. In addition, the time courses of miRNA expressions and miRNA primary transcript remained to be addressed. The data in Fig. 1 expand our previous analysis by providing these results.

In regard to Fig. 7 (Fig. 8 in the revised manuscript), these data not only

indicate the correlation of miR-17-92 and K⁺ channels, but also the potential for potassium channel modulators, especially their combination, to compensate for the channels downregulation. We, therefore, consider that the data in Fig. 7 are important from the viewpoint of therapeutic potential for potassium channel modulators. For these reasons, we would like to keep these data in Figs. 1 and 8.

We moved Fig. 5a–c to page 9, lines 8–14, to be presented in order.

Responses to Reviewer #2

Major points

miR-17-92 expresses all types of DRG neurons. Nerveless, it reduced Kv currents in small, but not medium/large, DRG neurons. The behavioral study showed that mi-17-92 affected mechanical allodynia, but not thermal hyperalgesia. How does this happen? The statement “the potassium channels affected are reportedly implicated only in mechanical sensation” (page 9, line 196-198) is improper. DRG Kv1.2 overexpression or knockdown affected paw withdrawal latency to thermal stimulation (Fan L et al., Mol Pain, 2014; Zhao X et al., Nature Neurosci 2013; Li Z et al., Pain 2015). More detailed discussion should be provided.

In addition to medium/large DRG neurons, mechanical allodynia was reported to be mediated by small non-peptidergic C fibers in a neuropathic pain model (Sherrer et al., Cell 137;1148–1159, 2009). Consistent with this, our additional expression pattern data (Fig. 5c) indicates that miR-17-92 is mainly colocalized with potassium channels mediating only mechanical allodynia (K_v3.4, K_v4.3) in non-peptidergic DRG neurons that are P2X3-positive and substance P-negative. We added this discussion to the revised manuscript (page 16, lines 6–9).

As the reviewer indicated, K_v1.2 has been shown to be involved in thermal sensation as well. However, K_v1.2 is not predicted as an miR-17-92 target, and indeed the K_v1.2 expression was not decreased by miR-17-92 overexpression (Supplementary Fig. 9). For clarity, we revised the sentences (page 15, line 24 to page 16, line 2).

In addition, all changes in the channel expression givens are at the levels of mRNAs. The changes in the amounts of their proteins should be provided.

We investigated the protein expression changes and added these data to the revised manuscript (Fig. 7a; page 11, line 20 to page 12, line 2).

Minor points

Comment 1: The accession number of Gene Expression Omnibus data used for bioinformatics analysis is missing.

We added the accession number of Gene Expression Omnibus data (GSE24982; page 7, line 15).

Comment 2: Supplementary Table 1 is missing.

The size for Supplementary Table 1 is so large that the Excel file is uploaded separately from the manuscript. Please see the Excel file named Supplementary Table 1.

Comment 3: miR-17-92 was up-regulated in the DRG after SNL, but not after CFA-induced inflammation. Does this phenomenon occur in other neuropathic pain models or only in SNL model? Moreover, the authors showed this up-regulation in the L5 DRG after SNL. It is unknown if such up-regulation also occurs in the intact DRG (i.e. ipsilateral L4 DRG) and the ipsilateral L5 spinal cord.

To address the reviewer's comment, we quantified miR-17-92 expression levels in another neuropathic pain model, the spared nerve injury model, and found that miR-17-92 was also increased after the spared nerve injury (Supplementary Fig. 2a, page 5, lines 3–4).

In the intact ipsilateral L4 DRG and ipsilateral L5 dorsal spinal cord, miR-17-92 expression levels were unchanged. These data were added to the manuscript (Supplementary Fig. 2b,c, page 5, lines 4–6).

Comment 4: DRG neurons are usually divided into three categories: small (< 600 μm^2), medium (600-1,200 μm^2) and large (> 1,200 μm^2). Some medium-sized DRG neurons are c-fiber or A δ -fiber ones.

In accordance with the reviewer's comment, we corrected the descriptions of "C-fiber" and "A-fiber" to "small-sized" and "medium/large-sized", respectively.

Comment 5: AAV usually requires several (at least 2-3 weeks) weeks to become expressed in vivo. It is interesting that the authors reported an increase in miR-17-92 at 7 days after AAV injection. There is no time course in the changes of its expression and behavioral responses after AAV injection.

As the reviewer suggested, the AAV vector may require several weeks to induce full expression of the transgene. Although a detailed time course analysis was not performed, we previously reported that the AAV vector can work at detectable levels at earlier time points (Sakai et al., Brain 2013).

Comment 6: A total of 5 ul AAV was injected with a 27-gauge microsyringe. Such volume and microsyringe may damage DRG cells significantly.

As the reviewer suggested, AAV injection may cause some cellular damage. However, our microinjection procedure limited the damage, and control AAV injection alone did not induce pain. To exclude a possibility of methodological bias, we always adopt control AAV injection as a control. We added the detailed microinjection method (page 22, line 24 to page 23, line 4).

Comment 7: Does the altering miR-17-92 expression affect motor function?

Altered miR-17-92 expression using an AAV vector was induced only in DRG neurons, not motoneurons. Neither miR-17-92 overexpression nor inhibition caused apparent motor dysfunction. No change in thermal withdrawal latency by miR-17-92 also supports a lack of apparent motor dysfunction. We have added the brief description to the Materials and Methods (page 23, lines 7–9).

Comment 8: In addition to mechanical allodynia, other typical neuropathic symptoms are cold allodynia and spontaneous ongoing pain. Does the miR-17-92 affect those symptoms?

We did not examine cold allodynia in the present study. We added the following discussion of the possible involvement of miR-17-92 in cold allodynia to the revised manuscript (page 16, lines 9–14): “Cold temperatures strongly and preferentially inhibit A-type currents but have fewer inhibitory effects on tetrodotoxin-resistant Na⁺ channels and non-inactivating K⁺ currents in small DRG neurons (Sarria et al., 2012), suggesting a contribution of A-type K⁺ currents to cold pain. Therefore, miR-17-92 may also have a modulatory role in nociceptive cold sensation.”

Apparent spontaneous ongoing pain behaviours, such as licking and flinching, are rarely observed in neuropathic pain models. Spontaneous ongoing pain has not been explicitly examined in most of the previous studies, and molecular mechanisms of spontaneous ongoing pain associated with neuropathic pain remain largely speculative. Therefore, we did not examine spontaneous ongoing pain behaviours and do not have any convincing evidence for a role of miR-17-92 in spontaneous ongoing pain associated with neuropathic pain.

Comment 9: The single DRG neuron RT-PCR results showed co-expression of multiple potassium genes and miR-17-92 cluster. It is rough to measure the size of the neurons to

classify them as C-fiber or A-fiber. Which internal control was used as the indicator of successful isolation of RNA from each neuron by LCM? miRNA in situ hybridization and immunofluorescence experiments will further to support these conclusions.

In accordance with the reviewer's comment, we corrected the descriptions of "C-fiber" and "A-fiber" to "small-sized" and "medium/large-sized", respectively. As an internal control, we assessed GAPDH expression, which was positively detected in all single DRG neurons (Fig. 5c). Because in situ hybridization for miR-18a and miR-19b did not work well, possibly because of its short nucleotide sequence, we performed RT-PCR of microdissection samples to assess expression of peptidergic and non-peptidergic DRG neuronal markers, TAC1 (encoding substance P) and P2X3 (Fig. 5c; page 9, lines 16–20).

Comment 10: The method of cell culture is missing

We added the method of cell culture (page 24, lines 2–4).

Comment 11: The detailed information of primers and the catalog number of qPCR kit are missing.

We added the detailed information of the qPCR kit (Supplementary Table 2; page 19, line 22 to page 20, line 1).

Comment 12: Please indicate the sequences of TuD RNA against individual miRNA or relevant catalog numbers in the Methods part.

We added the catalog numbers of TuD RNAs in the Methods (page 21, lines 9–12).

Comment 13: Some unnecessary open circles are found in Fig 2b and 4b.

In the "Reporting Checklist For Nature Communications Life Sciences Articles", it is recommended to plot individual data points for small sample sizes. Therefore, we would like to leave the open circles in the figures.

Responses to Reviewer #3

Comment 1: The study for acute pain, instead of Day 3, is required to understand the late phase of such epigenetic modulation in the development of neuropathic pain sensation.

In accordance with the reviewer's suggestions, we added the miR-17-92 expression data at Day 1 (Fig. 1a,c, page 4, lines 19–23) and found that miR-17-92 expression was

upregulated even 1 day after nerve injury.

Comment 2: The authors have nicely done the reporter gene assay. However, instead of RT-PCR, the western blotting assay should be required to determine the influence of miRNAs on targeted mRNAs. Simple comparison analysis in the complex modulation to target mRNAs/protein synthesis by miRNAs between RT-PCR and western blotting might be informative.

To address the reviewer's comment, we investigated the effects of antisense RNAs on K⁺ channel protein expressions and found that concurrent injection of AAV vectors expressing miR-18a, miR-19a, miR-19b and miR-92a blocked the protein expressions (Fig. 7a; page 11, line 20 to page 12, line 2).

Comment 3: A simple single cell analysis is unique and interesting. However, the present information of "cell heterogeneity" may not be enough to identify the cell types (ex: peptidergic or non-peptidergic cell). The authors should identify each cell type.

To address the reviewer's comment, we performed RT-PCR of microdissection samples to assess expression of peptidergic and non-peptidergic DRG neuronal markers, TAC1 (encoding substance P) and P2X3, respectively. miR-17-92 was primarily localized to P2X3-expressing non-peptidergic DRG neurons, although it was also expressed in the TAC1-positive peptidergic DRG neurons (Fig. 5c, page 9, lines 16–20).

Reviewers' comments:

Reviewer #1 (Remarks to the Author):

The authors have collected additional data to address several major concerns raised previously. Although the paper has improved somewhat, some issues were not fully addressed.

1. Although the authors tested some additional K⁺ expression, this analysis is still incomplete. As stated previously, the authors should use an unbiased microarray or RNA sequencing to show the gene profile (K⁺ channel genes and non K⁺ channel genes) altered by miR-17-92 overexpression or knockdown in the DRG. Without this information, it is unclear whether the effect of miR-17-92 on neuropathic pain is mediated SPECIFICALLY by the only 7 K⁺ channels in the DRG.
2. As to the discussion about the link of G9a to miR-17-92, there is no direct evidence showing that G9a (a histone modifier) inhibits transcription through DNA methylation. In fact, genome-wide analysis has shown that DNA methylation changes in the nervous system are either absent or very minor in various conditions.

Reviewer #2 (Remarks to the Author):

Although the authors responded to some comments raised, it is still unclear how the miR-17-92 cluster contributes to neuropathic pain. The authors used AAV6 to deliver siRNA or miR-17-92 into DRG neurons. What sizes of DRG neurons were infected by AAV6? This result must be provided. Otherwise, the behavioral data cannot be interpreted.

Minor points:

1. The previous studies from Devor M group showed the ectopic discharges in Abeta afferents as a source of neuropathic pain. The work from John wood group revealed that Na(v)1.8-expressing neurons were essential for mechanical, cold, and inflammatory pain but not for neuropathic pain. The authors have to discuss these findings in the discussion part.
2. siRNA has the off-target effect.
3. Because a big volume of AAV was injected, no cell damage and inflammation should be provided.
4. Because of the potential off-targets caused by siRNAs, whether the motor function is changed is unknown.
5. Spontaneous pain is one of major neuropathic pain symptoms. The authors should provide the effects of miR-17-92 on the spontaneous ongoing pain.

Reviewer #3 (Remarks to the Author):

The authors have clearly addressed all comments and added new data. I have no more comments and requests.

Responses to Reviewer #1

Comment 1: Although the authors tested some additional K⁺ expression, this analysis is still incomplete. As stated previously, the authors should use an unbiased microarray or RNA sequencing to show the gene profile (K⁺ channel genes and non K⁺ channel genes) altered by miR-17-92 overexpression or knockdown in the DRG. Without this information, it is unclear whether the effect of miR-17-92 on neuropathic pain is mediated SPECIFICALLY by the only 7 K⁺ channels in the DRG.

To address the reviewer's comment, we performed microarray experiment in the DRG overexpressing miR-17-92. We confirmed that miR-17-92 decreased the expression of the seven target potassium channels. miR-17-92 was also shown to modulate the expression of other potassium channels (Supplementary Table 2) and many other non-potassium channel genes (Supplementary Table 3) (page 10, lines 1–7). These molecules may be involved in pain caused by miR-17-92 overexpression, although the extent of their contribution is unclear.

Comment 2: As to the discussion about the link of G9a to miR-17-92, there is no direct evidence showing that G9a (a histone modifier) inhibits transcription through DNA methylation. In fact, genome-wide analysis has shown that DNA methylation changes in the nervous system are either absent or very minor in various conditions.

We appreciate the Reviewer pointing out our mistake. We corrected the discussion as follows: "G9a inhibits transcription through *histone* methylation" (page 15, line 2).

Responses to Reviewer #2

Although the authors responded to some comments raised, it is still unclear how the miR-17-92 cluster contributes to neuropathic pain. The authors used AAV6 to deliver siRNA or miR-17-92 into DRG neurons. What sizes of DRG neurons were infected by AAV6? This result must be provided. Otherwise, the behavioral data cannot be interpreted.

To address this comment, we analyzed the size of DRG neurons transfected with AAV6 and found that AAV6 transduction occurred in DRG neurons of all sizes, as previously described (Towne et al., Mol. Pain 5:52 (2009)) (Supplementary Fig. 3a; page 5, line 19).

Minor points

Comment 1: The previous studies from Devor M group showed the ectopic discharges in Aβ afferents as a source of neuropathic pain. The work from John wood group revealed that Nav1.8-expressing neurons were essential for mechanical, cold, and inflammatory pain but not for neuropathic pain. The authors have to discuss these findings in the discussion part.

In accordance with the reviewer comment, we added the findings about ectopic discharges in Aβ afferents as a source of neuropathic pain and the essential role of Nav1.8-expressing neurons in neuropathic pain to the revised Discussion (page 16, lines 15–18).

Comment 2: siRNA has the off-target effect.

As the reviewer pointed out, the off-target effect is an important issue regarding the siRNA experiment. However, in this study we used RNA decoy (tough decoy RNA) to prevent miRNA from binding to target mRNAs. Thus, RNA decoy, which differs from siRNA, blocks mRNA function by binding to miRNA, not by cleaving mRNAs. In fact, as described in our response to Comment 4 (see below), decoy RNA did not affect motor function. We added this description of tough decoy RNA to the revised Results section (page 6, lines 10–11).

Comment 3: Because a big volume of AAV was injected, no cell damage and inflammation should be provided.

We do not think that our microinjection maneuver is free from cell damage or inflammation, as is the case for other microinjection maneuvers. However, AAV injection alone did not induce pain behaviors as shown in the Results, suggesting that AAV injection-induced cell damage and inflammation, if any, does not interfere with the present experiment. We also performed open field and rotarod tests to validate the effect of the AAV injection on motor performance in the revised manuscript, and found that it did not affect motor function (page 7, lines 1–2). Furthermore, because we included a control AAV injection in all experiments, we believe that the effects of transgenes observed in the present study were certainly mediated by those encoded by AAV.

Comment 4: Because of the potential off-targets caused by siRNAs, whether the motor function is changed is unknown.

To address the reviewer's comment, we performed open field and rotarod tests after injecting AAV expressing tough decoy RNA against miRNAs. We found that neither the AAV injection itself nor decoy RNA expression caused motor dysfunction (page 7, lines 1–2).

Comment 5: Spontaneous pain is one of major neuropathic pain symptoms. The authors should provide the effects of miR-17-92 on the spontaneous ongoing pain.

To address the reviewer's comment, we assessed the effects of miR-17-92 inhibition on spontaneous pain by adopting a combined neuropathic (SNL) and inflammatory (complete Freund's adjuvant injection into the paw) pain model (Allchorne et al., *Neurosci. Res.* 74:230-238 (2012)), represented by obvious spontaneous foot lifting. Although axotomy of sciatic and saphenous nerves induces autotomy of digits as a possible sign of spontaneous pain, this model has ethical problems as noted by Koplovitch et al. (*Exp. Neurol.* 236:103-211 (2012)). In addition, for transducing AAV, removal of vertebrae overlying all L4–L6 DRGs which contributed to the injured nerve caused severe damage to rats in a preliminary study.

In the combined neuropathic and inflammatory pain model, miR-17-92 inhibition did not significantly suppress spontaneous paw lifting, although it had a tendency to suppress spontaneous pain behavior (Supplementary Fig. 5d; page 7, lines 2–7). This may be explained as the inflammatory component being less affected by miR17-92, consistent with the results in CFA model rats where miR-17-92 expression was not decreased (Supplementary Fig. 2d).

REVIEWERS' COMMENTS:

Reviewer #1 (Remarks to the Author):

The authors have provided additional microarray data showing K⁺ channel genes and other non-K⁺ channel genes affected by miR-17-92. With this information, I think that these analyses are complete and satisfactorily address my previous concerns.

Reviewer #2 (Remarks to the Author):

The Authors responded to most of the comments, but the response to spontaneous pain is improper. It would be better to use the CPP test. In addition, the images of AAV6-EGFP showing the viral distribution in the DRG are needed. It is surprised that DRG tissue damage caused by large volume injection did not produce behavioral changes. The images of the damage size need to be provided.

Responses to Reviewer #2

The Authors responded to most of the comments, but the response to spontaneous pain is improper. It would be better to use the CPP test. In addition, the images of AAV6-EGFP showing the viral distribution in the DRG are needed. It is surprised that DRG tissue damage caused by large volume injection did not produce behavioral changes. The images of the damage size need to be provided.

Image of DRG transfected with AAV6-miR-17-92-EGFP had been presented in the Figure 2a. In accordance with the editor's and reviewer's comments, we added the image of DRG transfected with AAV6-EGFP (Supplementary Fig. 3b).